# A role for brassinosteroid signalling in decision-making processes in the Arabidopsis seedling

**Nils Kalbfuß**[1], **Alexander Strohmayr** [1], **Marcel Kegel**[1], **Lien Le**[1], **Friederike Grosse-Holz**[1], **Barbara Brunschweiger** [1], **Katharina Stöckl**[1], **Christian Wiese** [1], **Carina Franke**[1], **Caroline Schiestl**[1], **Sophia Prem**[1], **Shuyao Sha** [1], **Katrin Franz-Oberdorf**[1], **Juliane Hafermann**[1], **Marc Thiemé**[1], **Eva Facher**[2], **Wojciech Palubicki**[3], **Cordelia Bolle**[4], **Farhah F. Assaad** [1]*

**1** Botany, School of Life Sciences, Technische Universität München, Freising, Germany, **2** Systematic Botany and Mycology, Faculty of Biology, Ludwig-Maximilians-University, Munich, Germany, **3** Mathematics and Computer Science, Adam Mickiewicz University, Poznań, Polen, **4** Plant Molecular Biology (Botany), Ludwig-Maximilians-University Munich, Martinsried, Germany

* farhah.assaad@tum.de

**Data Availability Statement:** All relevant data are within the manuscript and its Supporting Information files.

## Abstract

Plants often adapt to adverse conditions via differential growth, whereby limited resources are discriminately allocated to optimize the growth of one organ at the expense of another. Little is known about the decision-making processes that underly differential growth. In this study, we developed a screen to identify decision making mutants by deploying two tools that have been used in decision theory: a well-defined yet limited budget, as well as conflict-of-interest scenarios. A forward genetic screen that combined light and water withdrawal was carried out. This identified *BRASSINOSTEROID INSENSITIVE 2 (BIN2)* alleles as decision mutants with "confused" phenotypes. An assessment of organ and cell length suggested that hypocotyl elongation occurred predominantly via cellular elongation. In contrast, root growth appeared to be regulated by a combination of cell division and cell elongation or exit from the meristem. Gain- or loss- of function *bin2* mutants were most severely impaired in their ability to adjust cell geometry in the hypocotyl or cell elongation as a function of distance from the quiescent centre in the root tips. This study describes a novel paradigm for root growth under limiting conditions, which depends not only on hypocotyl-versus-root trade-offs in the allocation of limited resources, but also on an ability to deploy different strategies for root growth in response to multiple stress conditions.

## Author summary

The ability to grow in response to limiting, adverse conditions is a survival strategy unique to plants. This study addresses the tight regulation of differential growth in plants, and how growth is achieved under multiple stress conditions, in the absence of a carbon or energy source. We design a screen to identify decision-making mutants by deploying two tools that have been used in decision theory: a well-defined yet limited budget, as well as

**Funding:** This work was supported by Deutsche Forschungsgemeinschaft DFG grants AS110/4 and AS110/8-1 to F.F.A. and DFG grant BO1146/7-1 to C.B.; N.K. was funded by AS110/4 and A.S., B.B., S.P., S.S. and M.T. were funded by AS110/8-1. TUMmesa was funded with support of the German Science Foundation (DFG, INST 95/1184-1 FUGG). The funders had no role in study design, data collection and analysis, decision to publish, or preparation of the manuscript.

**Competing interests:** The authors declare that they have no conflict of interest.

conflict-of-interest scenarios. We also apply a new combination of stress factors: water stress and light deprivation. Our manuscript addresses tradeoffs in hypocotyl versus root growth, with an emphasis not on growth arrest but rather on enhanced growth responses to abiotic stress factors. Our findings challenge the widely accepted view that root growth correlates with meristem size; we propose alternative root growth strategies for adaptation to adverse, limiting conditions. Our genetic screens have identified the BIN2 clade of shaggy-like kinases as playing a central role in decision-making in the seedling. We address the controversy in the literature regarding brassinosteroid function at a cellular level. This we do by exploring the role of BR signalling in the regulation of cell elongation and geometry in response to abiotic stress cues. Our findings suggest that BIN2 and its homologues are required for the decision as to which growth strategy to adopt under different environmental conditions.

## Introduction

Plants often adapt to adverse conditions via differential growth, whereby limited resources are differentially allocated to optimize the growth of one organ at the expense of another. A good example of this is the etiolation response [1], in which shoot growth is prioritized over root growth in the dark. Another example pertains to changes in root architecture in response to phosphate, nitrate or water deprivation (reviewed in [2,3]). Differential growth is also a key feature of plant responses to environmental stimulus when resources are sufficient, as seen in phototropism or gravitropism. In the case of phototropism, shoot curvature is achieved by differential growth within the stem, with one side growing faster than the other [4]. In addition to differential growth decisions, plants need to assess trade-offs in the allocation of resources to defence versus growth [5–10]. Plants also make choices as to when to initiate developmental processes or transitions such as germination, bud emergence, fruit set or leaf drop as well as the switch from vegetative to reproductive growth. The timing of floral transitions is impacted by environmental cues such as day length and temperature. Thus, milder winters are currently giving rise to earlier flowering in temperate climates. Furthermore, with changing climate the cues that guide decision making have become more erratic; in some cases, these cues even appear contradictory, as in the case of mild winters followed by late frosts or of drought followed by flooding. Understanding decision making in plants and how such processes respond to erratic or contradictory cues therefore becomes imperative to an understanding of the impact of changing climate.

In a judgement and decision-making model for plant behaviour, judgement is described as consisting of discrimination, assessment, recognition and categorization, whereas decision making involves an evaluation of the costs and benefits of alternative actions [2]. Whether judgement and decision-making are empirically distinguishable remains to be determined. Nonetheless, it is interesting to reconsider the above-mentioned example of the floral transition within the framework of the judgement and decision-making model. When flowering occurs prematurely in a winter month, the questions that arise are what assumptions the plant can make as to how the spring will progress, as to when late frosts might set in, or as to the availability of water in the summer months to ensure a proper development of its fruit. It is not clear in this case what assumptions can be made and what degrees of uncertainty computed. Thus, in the formal language of decision theory [11], how a plant can assess the "state of the world" in the context of the floral transition appears unclear. The absence of such an assessment likely obscures the judgement required to inform decision-making. In brief, the

floral transition might be classified as a choice made under considerable degrees of uncertainty. Simpler and clearer decision problems include the allocation of—or competition for–limiting resources [12]. Competition for light between neighbouring plants has, for example, been studied in *Potentilla reptans*, which was shown to adopt one of three strategies–vertical growth, shade tolerance or lateral avoidance–to optimize above ground responses to prevailing light-competition scenarios [13]. Below ground, responses to variance in nutrient supply has been studied in split root pea exposed to variable and constant patches of soil under high or low mean nutrient concentrations; more roots developed in the variable patch when mean nutrients were low, whereas more roots developed in the constant patch when nutrients were high [14]. This example depicts a clear assessment of risk and a consistently preferred outcome depending on mean nutrient levels: plants were risk adverse under high nutrient supply but risk prone when nutrients were low. While this example enables us to visualize decision-making in plants, the underlying regulatory networks remain poorly understood.

In this study, we explored a variety of screen conditions designed to best visualize decision-making in Arabidopsis. To identify major players, we performed a forward genetic screen. This calls for a simple decision problem. Therefore, we considered decisions reached under limiting conditions, as well as conflict of interest scenarios. We focused on the germinating seed, which has a limited energy budget clearly defined as the nutrient, oil and protein body reserves available in the Arabidopsis endosperm and embryo. Before the seed's resources run out, the seedling must establish a root system capable of foraging for water and nutrients, and a photosynthetically active shoot system. Thus, the germinating seedling reaches binary shoot versus root growth decisions in terms of allocating the limited energy resources contained in the seed. We developed "conflict of interest" scenarios to monitor trade-offs between shoot versus root growth in the Arabidopsis seedling. These scenarios combine two abiotic stress factors that promote either hypocotyl or root growth. The ability to grow in response to limiting, adverse conditions is a survival strategy unique to plants; alternative responses adopted by yeast or animal cells include quiescence or the activation of apoptosis [15]. Our forward genetic screen identified BR signalling as playing a central role in decision-making in the seedling. We explore the strategies adopted to enable growth responses to abiotic stress cues within a limited budget, as well as the role of BR signalling in the deployment of such growth strategies.

## Materials and methods

### Lines and growth conditions

*Arabidopsis thaliana* lines used in this study are listed in **S1 Table**. Mutant lines were selected via the TAIR and NASC web sites [16]. EMS mutagenesis was carried out on Landsberg erecta (Ler) seed as described [17]. Seed were surface sterilized, stratified at 4˚C, and sown on Murashige and Skoog (MS) medium supplemented with B5 Vitamins (Merck group; https://www.sigmaaldrich.com). For nutrient stress conditions, NPK media was used. Plates were incubated under controlled growth chamber conditions (22˚C, 80 µmol m$^{-2}$s$^{-1}$). 10-day-old plate grown seedlings were used for organ length measurements and for scanning electron microscopy, and six or seven-day-old root tips were used for light microscopy. See **S1 and S2 Methods**.

### Screen conditions

For all screen conditions, seed were germinated on nutrient or nutrient-deficient medium without a carbon source (see **S1 Fig**). Water deficit plates were prepared with PEG-6000 (Merck group; https://www.sigmaaldrich.com) as described in **S3 Method**. Plants were grown under optimal conditions (16: 8 hr light: dark photoperiod, 22˚C, 180 µmol m$^{-2}$s$^{-1}$) at the

TUMmesa ecotron [18] and the sterilized seed were imbibed at 4˚C in the dark for 7 days (prior to plating) to break dormancy. Initial screen conditions compared growth on full-strength MS medium, with or without PEG-6000 for an additional -0.4 MPa. As the nutrients in MS medium generate a pressure of -0.2 MPa, initial screen conditions thus compared -0.2 MPa and -0.6 MPa. For optimized screen conditions, we later replaced full-strength MS by half-strength (½) MS medium, which corresponds to 0 MPa; thus, optimized screen conditions compare 0 MPa to -0.4 MPa. Under both initial and optimized screen conditions only the roots were exposed to the medium and, thus, to water stress; this was initially achieved via cutting a window in the agar and later via a plastic strip between the medium and the shoots. A final optimization was to incline the plates to promote root growth on the surface of the agar as opposed to in the agar. In the tables and graphs, we refer to initial versus optimized screen conditions to most accurately describe how the measurements were carried out. For growth under different light qualities blue, red or far-red LEDs (Quantum Devices) were employed. See **S3**–**S5 Methods**.

## Molecular methods

Mapping, positional cloning and allele sequencing were carried out as described in Jaber et al., 2010 [19] using primers tabulated in **S6 Method**. qPCR was carried out as described in **S7 Method**.

## Light and electron microscopy

For scanning electron microscopy, a Zeiss (LEO) VP 438 microscope was operated at 15 kV. Fresh seedlings were placed onto stubs and examined immediately in low vacuum. Confocal microscopes used for imaging were an Olympus (www.olympus-ims.com) Fluoview 1000 confocal laser scanning microscope (CSLM) and a Leica (www.leica-microsystems.com) SP8 Hyvolution CSLM. 40x and 60x water immersion 0.9 numerical aperture objectives (Olympus) were used. FM4-64 staining was as described in Ravikumar et al. [20], with emission at 640 nm. Root apical meristem properties are described in **S8 Method.**

For GUS staining, seed were plated under initial screen conditions (MS salts) and 7- or 10-day-old seedlings were used. Seedlings were fixed for 20 min in 90% acetone, washed with staining buffer (10 mM EDTA, 0.1% Triton-X 100, 2 mM potassium ferrocyanide, 2mM potassium ferricyanide, 50 mM sodium-phosphate buffer (pH 7)) on ice and transferred into staining buffer containing additionally X-Gluc (2.31 mM) and chloramphenicol (100 mg/ml). Seedlings were incubated at 400 mbar for 15 min on ice before being incubated for 3–5 h at 37˚C. Seedlings were dehydrated in an ethanol series (20%, 35%, 50%, 30 min per concentration) and finally fixed for 30 min at room temperature in FAA (50% ethanol, 3.7% formaldehyde, 5% acetic acid). Micrographs were recorded with an Olympus BX61 microscope. A 40x objective (water) was used.

## Image processing, data and statistical analysis

Shoot (hypocotyl) and root lengths were scored with the Image J free-hand tool (https://imagej.nih.gov). Images were processed with ImageJ, Adobe photoshop (www.adobe.com) and assembled with Adobe Illustrator. The hypocotyl volume was computed by assuming a cylindrical organ shape as follows: $V = \pi WL$, where W is the width and L the length of the hypocotyl.

Root and hypocotyl responses to water deficit in the dark (abbreviated as darkW) were computed as

$$\frac{\text{organ length in the dark}}{\text{organ length under darkW}}$$

And for the ratio:

$$\text{ratio adjustment} = \frac{\left(\frac{\text{hypocotyl}}{\text{root}}\right)\text{dark}}{\left(\frac{\text{hypocotyl}}{\text{root}}\right)\text{darkW}}$$

Due to variability between PEG lots and PEG plates (see text), we normalized each mutant to the corresponding wild-type ecotype (see **S1 Table**) on the same plate. Thus, the normalized ratio adjustment to water stress in the dark (darkW) was computed as follows:

$$\text{RQ}_{\text{ratio}} = \frac{\text{ratio adjustment mutant}}{\text{ratio adjustment wild type}}$$

$\text{RQ}_{\text{hypocotyl}}$ and $\text{RQ}_{\text{root}}$ as well as light versus dark responses were computed in a similar fashion. The mean $\text{RQ}_{\text{ratio}}$ of at least three biological replicates (i.e. the seed stocks from different mother plants) is shown. Responses were considered to be attenuated for $\text{RQ}_{\text{ratio}} < 0.8$, normal in the 0.8–1.2 range, and exaggerated for $\text{RQ}_{\text{ratio}} > 1.2$. For volcano plots we plotted the mean $\text{RQ}_{\text{ratio}}$ on the $X$-axis and the median $P_{ratio}$-value on the $Y$-axis. $P$-values were computed with the Student's $T$-test when the two populations had equal variances and with the Welch's $T$-test when variances were unequal. That distributions were normal was verified with the Shapiro-Wilk test for selected samples, and in the rare cases where normality was not clear, we applied the non-parametric Mann-Whitney-U tests. The computations were carried out in excel for graphic rendition and verified in R. For multiple testing, as carried out when different wavelengths and light intensities were compared, we applied a Benjamini–Hochberg correction. This is specified in the legend where applicable. Responses were considered to be insignificant for $P$-values $\geq 0.05$ and attenuated for $P$-values $\geq 0.00001$. The median $P$-value for at least three replicates is shown in the volcano plots.

## Results

### Hypocotyl growth in search of light is prioritized over root growth in search of nutrients in the young seedling

To understand decision making processes in the Arabidopsis seedling, we set up a "conflict of interest" scenario between shoot and root growth. To this end, seed were germinated under growth conditions designed to place contradictory demands on hypocotyl (growth at low light intensities or in the dark) or root growth (nutrient deficiency or water stress). We first tested several nutrient media for their ability to promote trade-offs between hypocotyl versus primary root growth in dark-grown seedlings; these include -P, -N, and -K media lacking phosphate, nitrate, or potassium (**S1 Fig**). We also tested low levels of osmotic stress (100 mM mannitol; **S2 Fig**) as well as salt stress (50–100 mM NaCl; **S2 Fig**), which has been shown to impair hypocotyl elongation in response to far-red light [21]. None of these media, however, gave rise to considerable or reproducible trade-offs between root and hypocotyl growth, by which we refer to the growth of one organ at the expense of another (**S1 and S2 Figs**). There was a clear priority for hypocotyl growth in search of light over primary root growth in search of nutrients (**S1D Fig**). It is to be noted in this context that we are looking not at root architecture but

exclusively at primary root growth and that the seed and embryo contain sufficient nutrients to support the initial phases of seedling growth [22].

## Opposing gradients of water stress and light intensity enable us to visualize shoot versus root growth trade-offs in the seedling

We next attempted to identify what resources might be as important to a germinating seedling as the light. Here, we explored the relative importance of water. To this end, we germinated seedlings in the presence of water stress, using polyethylene glycol (PEG) to withdraw available water from the medium in a standardized tissue culture setting [23,24]. In the light, PEG significantly decreased hypocotyl growth ($P_{hypocotyl}$ = 3E$^{-48}$), but this was not mirrored by an increase in root growth; root growth was, in fact, slightly reduced ($P_{root}$ = 1.3E$^{-02}$; **S3A Fig**). In the dark, increasing degrees of water stress considerably and reproducibly increased root length and decreased hypocotyl length (three-fold change for the hypocotyl and almost four-fold change for the root for -0.7 MPa compared to -0.2 MPa; **Figs 1A–1E** and **S3B–S3D**). In the light, water stress decreased the total length of the seedlings ($P_{total}$ = 8.2E$^{-4}$; **S3A Fig**). In the dark, however, water stress did not impact the total length of the seedling (for $\leq$ -0.6 MPa compared to -0.2; **Figs 1E** cf. **S3A**). In comparison to water-stress in the light, we were clearly observing trade-offs between hypocotyl and root growth in response to a gradient of water stress in dark-grown seedlings. We then tested a gradient of decreasing light intensity. Under low light, the trade-off was more pronounced, with longer roots and shorter hypocotyls than in the dark, even when the light intensity was as low as 2 µmol m$^{-2}$ s$^{-1}$ (**Fig 1F–1I**). Similarly, for both blue and red light, hypocotyl length increased whereas root length decreased with decreasing light intensity and this trend was enhanced by water stress (-0.6 MPa; **S4A and S4B Fig**). Far-red light showed a comparable trend (**S4C Fig**). In conclusion, using opposite gradients of decreasing light intensity versus increasing water stress, we were able to fine tune root growth at the expense of hypocotyl growth.

We next asked whether the reduction in hypocotyl length observed in response to water stress applied in the dark was accompanied by a change in organ width and, ultimately, volume. To this end, seedlings were imaged by scanning electron microscopy (**Fig 1A**) and the lengths and widths of the hypocotyls measured (**S5A and S5B Fig**). In a comparison between light to dark, the hypocotyl length and volume increased despite a decrease in hypocotyl width (**S5A–S5C Fig**). When water stress was applied to dark-germinated seedlings, the reduced length of the hypocotyl was not accompanied by an increase in width; in fact, a slight reduction in width was observed and overall the volume of the hypocotyl was two-fold reduced (**S5A–S5C Fig**). The change in organ volume suggests that resources such as water, which accounts for organ volume to a large extent, are being differentially allocated in response to light and water stress. In summary, hypocotyl and root lengths and the ratio thereof are exquisitely fine-tuned to the wavelength and intensity of the light source, and to the severity of water stress. In the case of the seedling, what we observe is a clear consistency in a continuum of preferred outcomes along a gradient in response to opposing gradients of light intensity and water stress.

## A forward genetic screen for "decision" mutants identifies BRASSINOSTEROID INSENSITIVE 2 (BIN2)

The fine-tuning of hypocotyl/root ratios can conceptually be broken down into four steps: (i) sensing, (ii) downstream signalling, (iii) decision making processes, and (iv) the execution of these decisions (the action). The gradients of hypocotyl and root growth we describe in response to our screen conditions can help distinguish between mutants impaired in the process of decision making *per se*, versus mutants with primary defects in one of the other steps

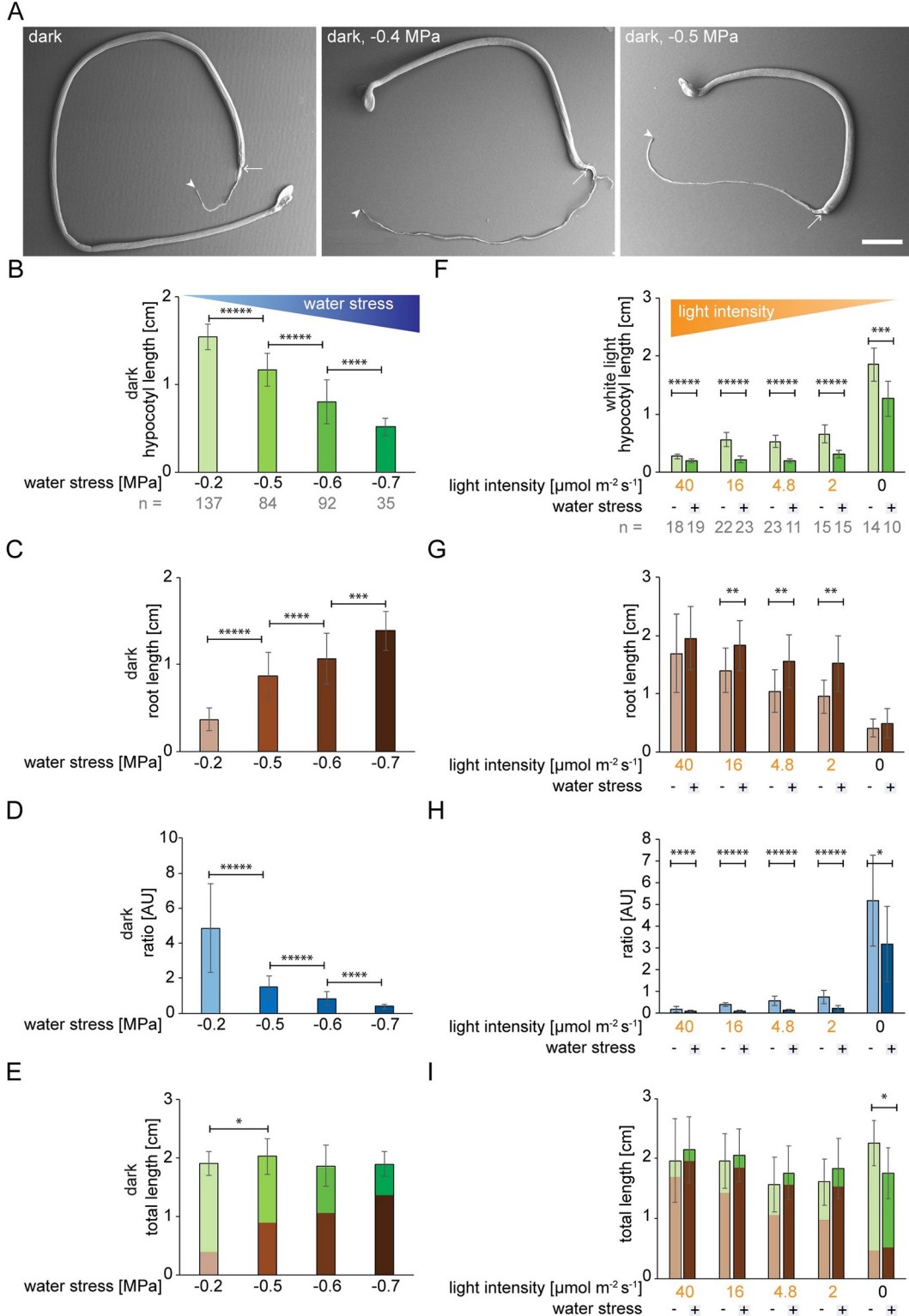

**Fig 1. Hypocotyl versus root growth in response to light and water availability.** Wild type (Col-0). (A-E) germination in the dark on a gradient of water stress ranging from 0 to -0.7 MPa. Recordings were taken 10 days after germination. (A) scanning electron micrographs; arrows point to the hypocotyl/root junction and arrowheads to the end of the root. (B-E) panels are from the same experiment. Water stress applied in the dark increases root length (C) at the expense of hypocotyl length (B), giving rise to a decrease in the hypocotyl/root ratio (D). Note that the total seedling length does not vary at -0.2

MPa versus -0.6 or -0.7 MPa (E). Thus, there was a clear tradeoff, by which we refer to the growth of one organ at the expense of another. (F-I) germination on MS medium under white light at varying intensities ranging from 40 to 0 μmol m$^{-2}$ s$^{-1}$, with or without -0.6 MPa water stress. A decreasing light intensity gradient increases hypocotyl length (F) at the expense of root length (G). The hypocotyl/root ratio was calculated (H). The number (n) of seedlings measured per condition is in grey below the mean ±StDev bar graphs. *P*-values were computed with a two-tailed student's *T*-test and are represented as follows: *: 0.05–0.01; **: 0.01–0.001; ***: 0.001–0.0001; ****: 0.0001–0.00001, *****: < 0.00001. See related **S1**, **S2**, **S3**, **S4** and **S5** **Figs**.

(sensing, signalling or execution). Perception mutants would fail to perceive light or water stress; a good example of this is the *cry1 cry2 phyA phyB* quadruple photoreceptor mutant [25], which had a severely impaired light response (**S4F Fig**), but a "normal" response to water stress in the dark (**S4G Fig**). In contrast, execution mutants may have aberrantly short hypocotyls or roots that are nonetheless capable of differentially increasing in length depending on the stress conditions. Decision mutants would differ from perception or execution mutants as they would clearly perceive the stress factors yet fail to adequately adjust their hypocotyl/root ratios in response to a gradient of multiple stress conditions. Failure to adjust organ lengths would be seen as a non-significant response, or as a significant response but in the wrong direction as compared to the wild type. We thus used organ lengths, the hypocotyl/root ratio and the significance of the responses as decision read outs. We specifically looked for mutants in which at least one organ exceeded wild-type length under darkW. To operate under limiting conditions, we germinated seedlings in the dark on media lacking a carbon source (see **S1 Fig**), avoiding even low levels of light.

To identify genes implicated in decision making, we performed a genetic screen in two consecutive steps (**Fig 2A**). In the first step, we germinated seed in the dark. In the second round, viable mutants with aberrant hypocotyl to root ratios in round 1 were rescreened for their ability to adjust their hypocotyl to root ratios in response to water stress in the dark. We initially screened in the dark because the high variance in root growth under water deficit in the dark in the wild type (see below) would obscure the distinction between putative mutants versus stochastically occurring wild-type seedlings with short roots under darkW. 83000 EMS-mutagenized M2 seed were subjected to the first screen and over 100 viable mutants rescreened in the dark in the presence or absence of water stress. 19 viable seedlings with aberrant hypocotyl to root ratios in our forward genetic screen had brassinosteroid-related dwarf phenotypes (**Figs 2B**, **S6A**, **and S6C**). Of these, one—named B1—had a very pronounced inability to adjust its hypocotyl/root ratio in response to multiple stress conditions: while the root response to water stress in the dark was only slightly attenuated ($P_{root}$ = 6E$^{-05}$; **Fig 2C**), the hypocotyl response was aberrant ($P_{hypocotyl}$ = 0.05; **Fig 2C**) and the ratio adjustment insignificant ($P_{ratio}$ = 0.83; **Fig 2C**). Positional cloning and allele sequencing suggested that the B1 locus encoded BRASSINOSTEROID-INSENSITIVE 2 (BIN2; **Fig 2D and 2E**). Our B1 allele carries a TREE domain E263K mutation, identical to *bin2-1* and shown to block the BR-induced ubiquitination and degradation of BIN2 [26]. Segregation analysis showed that our B1 *bin2* allele was semi dominant, like all TREE domain mutations of *bin2-1*. We sequenced 19 F2 individuals of the segregating mapping population and found that the E263K point mutation absolutely segregated with the phenotype (**S2 Table**). For complementation analysis, we crossed our B1 mutant with a known *bin2-1* allele [27]; F1 individuals had phenotypes on soil that were characteristic of homozygous *bin2* plants and, upon sequencing, exhibited EMS-induced G to A transitions at position 989 for both the B1 mutation and the *bin2-1* allele (**2E Fig**). We complemented B1 with *bin2-1* and *ucu1* alleles and compared it to *bin2-1*, *ucu1* and *dwarf12* [28,29] alleles at the BIN2 locus; these three published mutant lines exhibited the same behaviour as B1, including semi-dominance and partial etiolation. Indeed, the B1 phenotype was similar to *bin2-1* under our multiple stress conditions (**S6B Fig**; compare B1 in **Fig 2C** to *bin2-1* in

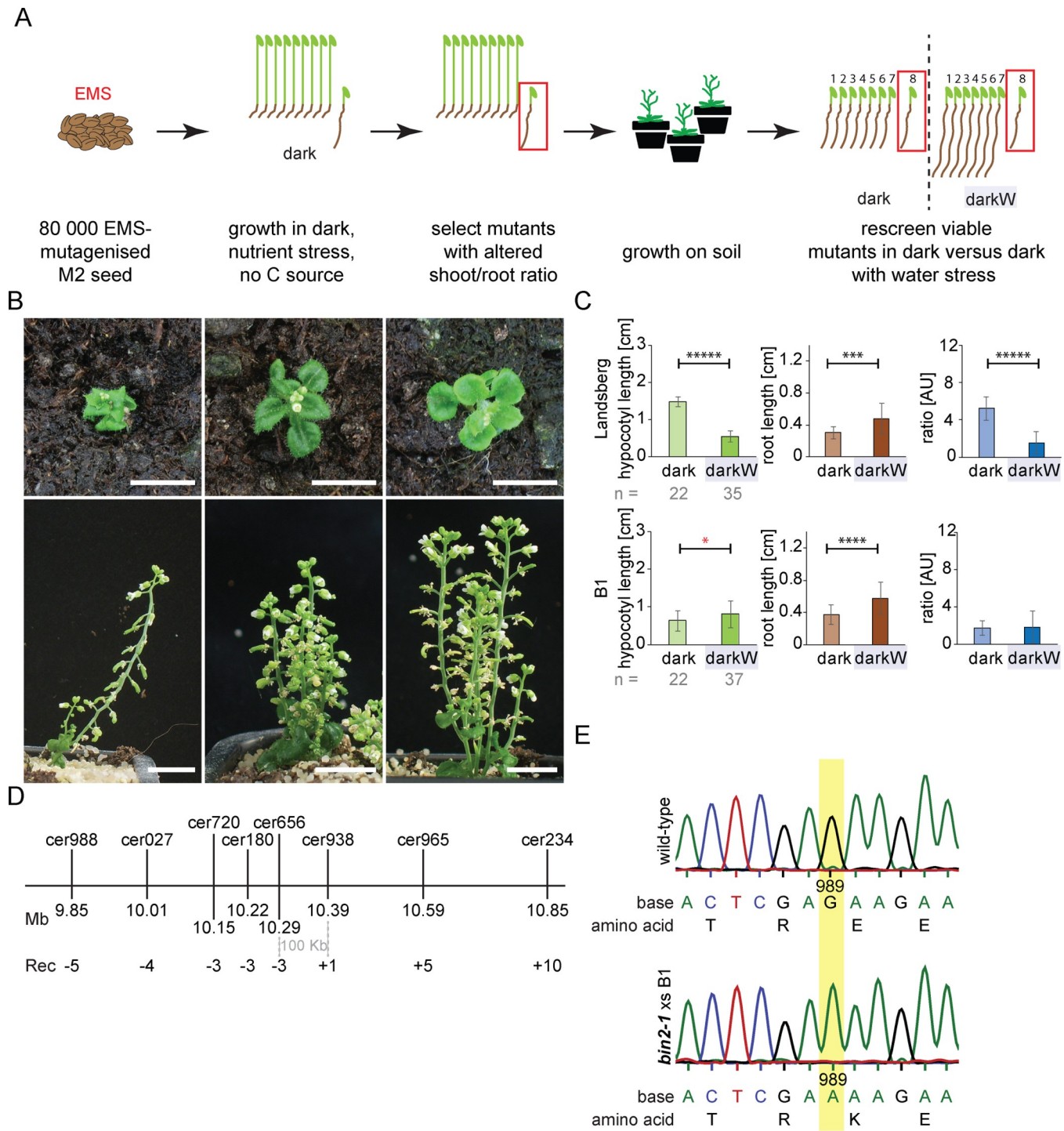

**Fig 2. Identification of BIN2 and its role in hypocotyl/root tradeoffs.** (A) A screen for decision mutants; two consecutive steps are depicted (see text). (B) Dwarf mutants with curled-in leaves, a BR-related phenotype, identified in our forward genetic screen on the basis of altered hypocotyl/root ratios in the dark. Scale bars = 1cm. (C) B1 mutants fail to adjust their hypocotyl/root ratios in response to water stress in the dark (darkW). Note that hypocotyls had an inverse response (significance marked with red asterisk). The number (n) of seedlings measured per condition is in grey below mean ±StDev bar graphs. *P*-values were computed with a two-tailed student's *T*-test and are represented as follows: *: 0.05–0.01; ***: 0.001–0.0001; ****: 0.0001–0.00001, *****: < 0.00001. (D) Fine mapping of mutant B1 identifies a 100 kb interval on chromosome 4, which spans the *BIN2* locus. Markers used for mapping are depicted in abbreviated form above the line and in full detail in **S6 Method**. Rec: recombinants. See S6 Method for mapping and **S2 Table** for segregation analyses. (E) Sequencing of the BIN2 TREE domain in F1 segregants of a complementation cross between *bin2-1* and B1 shows that B1 harbors a G to A transition at position 989 (yellow highlight), giving rise to an E263K mutation identical to that of *bin2-1*. n = 10 F1 plants were sequenced. See related **S6 Fig**.

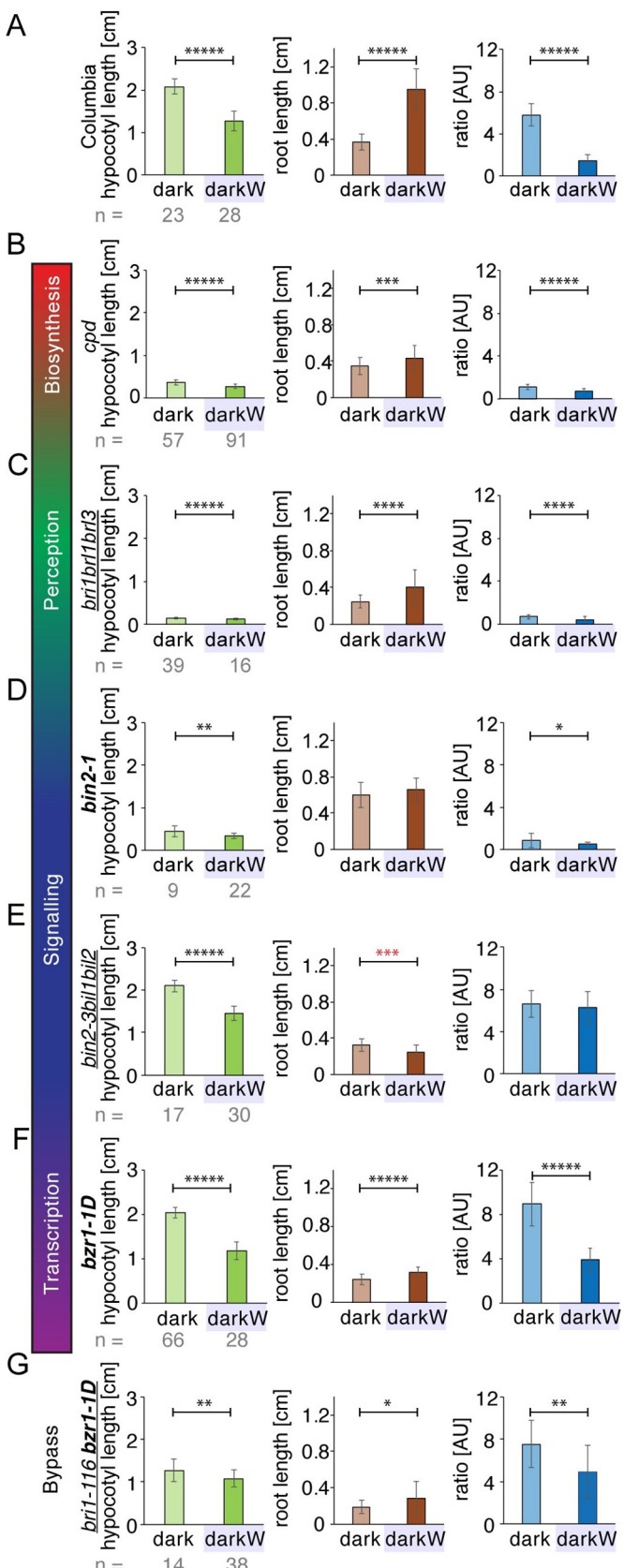

**Fig 3. Role of BR signalling in hypocotyl/root tradeoffs.** Seedlings were germinated on ½ MS in the dark (dark) or in dark with -0.4MPa water stress (darkW). (A) Col-0 (wild type). (B) BR biosynthesis mutant *cpd*. (C) BR perception mutant *bri1brl1brl3* (a segregating triple null). (D) BR signalling mutant *bin2-1* (a semidominant gain of function allele). (E) *bin2-3bil1bil2* tripple knockout; note that, in stark contrast to the wild type, the *bin2-3bil1bil2* tripple knock-out has shorter roots under darkW than in the dark (red asterisks). (F) Transcription factor mutant *bzr1-1D*, a dominant allele. (G) BIN2 bypass mutant *bri-116 bzr1-1D*. Note that *bin2-1* mutants have a severely attenuated hypocotyl response (D), *bin2-3bil1bil2* mutants have an inverse root response (red asterisks denote a significant response in the opposite direction to the wild type) and no ratio response (E), and *bri-116 bzr1-1D* severely attenuated hypocotyl, root and ratio responses (G). Null alleles are depicted in regular font, semi-dominant or dominant in bold and higher order mutants are underlined. At least 3 experiments were performed for each line, and a representative one is shown here on the basis of RQ and P values (see **Fig 4E and 4F**). The number (n) of seedlings measured per condition is in grey below the mean ±StDev bar graphs. *P*-values were computed with a two-tailed student's *T*-test and are represented as follows: *: 0.05–0.01; **: 0.01–0.001; ***: 0.001–0.0001; ****: 0.0001–0.00001, *****: < 0.00001. For mean RQ values and median *P*-values see **Figs 4E, 4F**, and **S8**. Mutant alleles and the corresponding ecotypes are described in **S1 Table**. See related **S6**, **S7**, **S8**, **S9**, **S10** and **S11 Figs**.

**Fig 3D**). In conclusion, positional cloning, allele sequencing, segregation analysis, complementation analysis and phenotypic analyses show that the B1 locus encodes BIN2.

## The BR pathway is implicated in hypocotyl versus root trade-offs in the Arabidopsis seedling

Brassinosteroids are known to be involved in the response to abiotic stress cues such as drought and salinity [21,30,31]. To analyse the role of BR signalling in decision making processes, we studied a set of known mutants impaired in BR biosynthesis, perception, signalling or in BR-responsive gene expression (**S1 Table**). In addition to bar graphs representing hypocotyl and root lengths (**Fig 3**), the distribution of datapoints was represented by violin plots (**Figs 4A–4C and S7**). The violin plots compare organ length distributions in mutants versus the corresponding wild-type ecotype, which depicts dwarfism in some brassinosteroid mutants. It is also apparent that wild-type (Col-0) root length varies under water-deficit in the dark (**S7 Fig**). Although we have optimized protocols for PEG plates to the best of our ability, there is still a lot-to-lot and plate-to-plate variation. This emphasizes the need for normalizing each mutant line to its corresponding wild-type ecotype on the same (PEG) plate in the same experiment. To this end, the response to water stress in the dark was represented as a normalized response quotient (RQ), which is an indication of how much the mutant deviates from the corresponding wild type (**Fig 4E**; see methods). We used the normalized ratio response $RQ_{ratio}$ as our main "decision" readout (**Fig 4E**) and show $RQ_{hypocotyl}$ and $RQ_{root}$ in the supplement (**S8 Fig**). This is because water stress in the dark is a "conflict of interest" scenario in which hypocotyl and root growth have competing interests (see **Fig 1**). We reason that mutants unable to integrate environmental cues might have a "confused phenotype" under our multiple stress conditions. The read out for a "confused phenotype" would translate into erratic (i.e. highly variable) hypocotyl versus root lengths. This high variance would, in turn, translate into a low signal to noise ratio, and this can be seen as a high *P*-value. We, therefore, plotted the median *P*-values against the normalized response quotients (referred to as volcano plots; mean $RQ_{ratio}$ in **Fig 4F**). Wild-type ecotypes invariably had low P values $< 10^{-10}$ (grey shading on the red line in **Fig 4F**, green arrow). Mutants with high *P*-values and low response quotients would be considered "confused" and these would map in the lower left quadrant of the $RQ_{ratio}$ volcano plot (response to water stress in the dark, **Fig 4F**).

In the BR pathway (**Fig 4D**), we first looked at *cpd*, a BR biosynthesis mutant. *CPD (CONSTITUTIVE PHOTOMORPHOGENIC DWARF)* encodes a cytochrome P450 monooxygenase involved in the C6 oxidation pathway of brassinolide biosynthesis [32]. *cpd* mutants are dwarfs with attenuated but significant responses to water stress in the dark (**Figs 3B, S7, 4E, 4F, and**

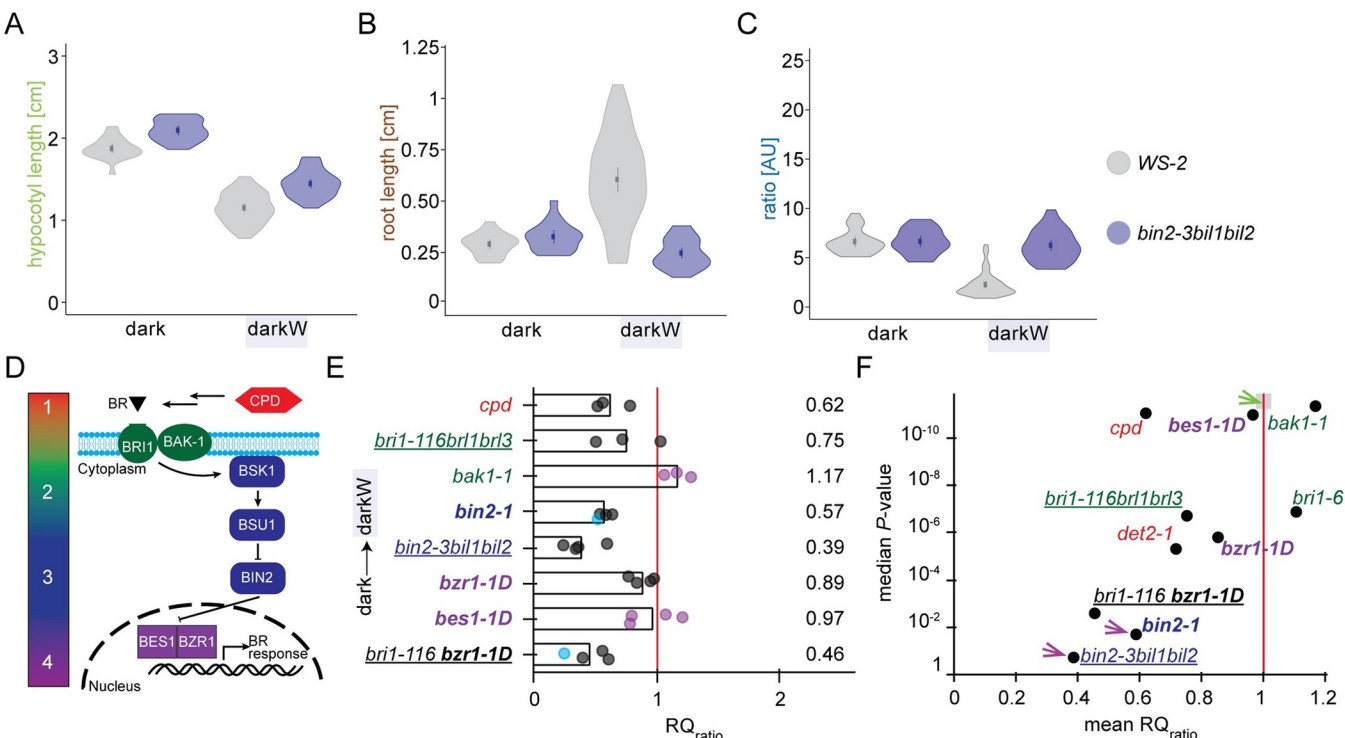

**Fig 4. Responses of BR pathway mutants to water stress in the dark: violin plots, response quotients and volcano plots.** (A-C) Violin plots of the hypocotyl (A), root (B) and ratio responses (C) of the triple *bin2-3bil1bil2* knock out line shown in **Fig 3E**, with the wild-type ecotype (Ws-2) as reference. The dot represents the mean and the line the 95% confidence interval. Note the high variance of the wild-type root response under darkW. *bin2-3bil1bil2* mutants qualified as decision mutants on 3 counts: (i) failure to adjust the hypocotyl/root ratio to darkW (the ratio for darkW is the same as for dark in panel C), (ii) low or non-significant P-value (see panel f below) and (iii) one organ (here the hypocotyl in panel a) exceeded wild-type length under darkW. (D) Color coding of steps in the BR signalling pathway; 1: biosynthesis, 2: perception, 3: signalling, 4: transcription. See text for further detail. (E) $RQ_{ratio}$ response quotient of the hypocotyl/root ratios under dark/darkW conditions, normalized to the wild-type ratio quotient; a value of 1 (vertical red line) indicates that the response to a shift from dark to darkW is similar to that of the respective wild-type ecotype. Each replicate is represented by a dot; purple dots are for initial and grey dots for optimized screen conditions; blue dots are for data from SEM measurements. Note that the triple *bin2bil1bil2* knock out has the strongest phenotype, followed by *bri1-116 bzr1-1D* and *bin2-1*. (F) Volcano plot with the mean $RQ_{ratio}$ depicted in (E) on the *X*-axis and the median *P*-Value of the response on the *Y*-axis (negative log scale; a median of all replicates was used). Mutants in the lower left quadrant are considered to have a "confused decision phenotype" (see text). Only *bin2* alleles (magenta arrows) or bypass mutants map to this quadrant. The area shaded in grey on the red line (green arrow) is where wild-type ecotypes would theoretically map onto the plot. Mutant alleles and their ecotypes are described in **S1 Table**. Null alleles are depicted in regular font, semi-dominant or dominant in bold and higher order mutants are underlined. See related **S6**, **S7**, **S8**, **S9**, **S10** and **S11 Figs**.

**S8**). We then turned to higher order mutants of the BR receptor, BRI1, and its homologues, *bri1-116 brl1 brl3* [33]. *bri1-116 brl1 brl3* mutants had an attenuated response to water stress in the dark; however, as for *cpd*, all trends were similar to the wild type and all responses significant (**Figs 3C, S7, 4E, 4F, and S8**). An allele of the BAK1 coreceptor, *bak1-1* [34], showed fairly similar responses to a hypomorphic *BRI1* allele, *bri1-6* (**Fig 4F**).

When BR binds to the extracellular domain of the BRI1 receptor, the BIN2 BRASSINOSTEROID INSENSISTIVE 2 shaggy-like kinase is inactivated (**Fig 4D**). We assessed the phenotype of the semi-dominant *bin2-1* allele, which has an identical amino acid change (TREE to TRKE) in the tree domain as our *bin2* B1 allele. For *bin2-1*, the response to water stress in the dark was severely impaired (**Figs 3D and S7**), with considerably attenuated hypocotyl and hypocotyl/root ratio responses (mean $RQ_{hypocotyl}$ = 0.65—see magenta arrow in **S8A Fig**; mean $RQ_{ratio}$ = 0.57 in **Fig 4E**; median $P_{ratio}$ = 0.02—see magenta arrow in **Fig 4F**). *bin2-1* seedlings are dwarfs with severe and pleiotropic phenotypes [27], including a reduction in seed size. We, therefore, turned to higher order null alleles. The Arabidopsis genome encodes ten shaggy-like kinases, of which

three in the BIN2 clade have been shown to function redundantly [35]. A triple knock-out of all three clade 2 shaggy like kinases, *bin2-3bil1bil2*, has a large stature, short roots and an overall phenotype characteristic of plants with enhanced BR signalling outputs [35]. Seed weight in *bin2-3bil1bil2* did not differ from the wild type (**S9A Fig**). This mutant was severely impacted, with a non-significant (median $P_{root}$ = 0.44; mean $RQ_{root}$ = 2.42 (magenta arrow in **S8B Fig**) or aberrant root response to water stress applied in the dark (**Figs 3E, 4E, and 4F**). *bin2-3bil1bil2* mutants fit the above definition of decision mutants as they have a significant root response but in the wrong direction as compared to the wild type, as denoted by red asterisks (**Fig 3E**). The hypocotyl/root ratio response was also non-significant (median $P_{ratio}$ = 0.19; magenta arrow in **Fig 4F**) and severely impaired (mean $RQ_{ratio}$ = 0.39; **Fig 4E**). Whereas the hypocotyl response was severely impaired in the semi-dominant *bin2-1* allele (magenta arrow in **S8A Fig**; cf. green arrow for *bin2-3bil1bil2*), the root response was severely impaired in the loss of function *bin2-3bil1bil2* line (magenta arrow in **S8B Fig**; cf. green arrow for *bin2-1*). This shows opposite phenotypes in gain- versus loss- of function *bin2* alleles. In brief, both semi-dominant and recessive mutations in clade 2 shaggy-like kinases impair differential growth responses to light and water stress in the seedling.

The active BIN2 kinase phosphorylates BZR and BES transcription factors, which reduces their DNA-binding activity, excludes them from the nucleus and targets them for degradation [36]. By inactivating BIN2, BR signalling results in the accumulation of unphosphorylated BZR1/2 in the nucleus and a concomitant expression of BR-target genes. We assessed the behaviour of a dominant BZR1 allele, *bzr1-1D* [37], which is, like *bin2-3bil1bil2*, characterized by enhanced BR signalling outputs [35]. Seed weight was not impacted in *bzr1-1D* (**S9B Fig**). With respect to water stress in the dark, the root and hypocotyl/root ratio responses were significant and fairly similar to the wild type (mean $RQ_{root}$ = 1.17; mean $RQ_{ratio}$ = 0.89: **Figs 3F, 4E, 4F, and S8B**). A BES1 dominant allele, *bes1-1D* [38], had a weaker phenotype than *bzr1-1D* in that it did not exhibit a considerable difference from the wild type in terms of its response to water stress in the dark (mean $RQ_{ratio}$ = 0.97; median $P_{ratio}$ = 2$E^{-11}$; **Fig 4E and 4F**).

Looking at BR pathway mutants suggests that BR signalling is implicated in hypocotyl versus root growth trade-offs in the Arabidopsis seedling. To address this hypothesis, we looked at the *bri1-116 bzr1-1D* double mutant [37], in which a null non-viable BRI1 receptor mutant is partially rescued by a dominant BZR1 transcription factor allele. In this line and downstream of BIN2, BZR1 is constitutively active regardless of the BR signal, which is effectively bypassed. *bri1-116 bzr1-1D* double mutants had a severely attenuated hypocotyl response to water stress in the dark (mean $RQ_{hypocotyl}$ = 0.58; median $P_{hypocotyl}$ = 0.01; **Fig 3G**; magenta arrow in **S8A Fig**). Similarly, the double mutants had very short roots in the dark that elongated erratically ($RQ_{root}$ = 1.43; non-significant median $P_{root}$ = 0.07; **S8B Fig**; note the large variance in **Fig 3G**) when water stress was applied in the dark. The hypocotyl to root ratio adjustment was severely impaired (mean $RQ_{ratio}$ = 0.46; median $P_{ratio}$ = 0.003; **Figs 3G, 4E, and 4F**). In summary, BR bypass mutants mapped together with *bin2* gain of function and loss of function mutants in the same quadrant of the volcano plots, showing a "confused" decision phenotype (high *P*-value, low ratio quotient; **Fig 4F**). We conclude that BR signalling and/or the BIN2 clade of shaggy-like kinases are required for differential growth decisions in the Arabidopsis seedling.

## BR pathway mutants perceive light and water withdrawal

We next asked whether BR mutants are capable of perceiving and responding to light or water-stress. We first compared light versus dark conditions. Some BR pathway mutants exhibited etiolated phenotypes (short hypocotyl, long root, at least partially open cotyledons in

the dark), as described in the literature [32]. Nonetheless, and even though responses were attenuated in some mutants, all BR pathway mutants had significant responses to light (shorter hypocotyls, longer roots as compared to dark-grown seedlings; **S10 and S11** Figs). *bin2-1* had a dwarf phenotype and an aberrant hypocotyl/root ratio, with a short hypocotyl and a relatively long root even in the dark (**S10D and S11** Figs). Interestingly, the BR mutant lines with the strongest etiolation phenotypes (*cpd* and *bri1-116brl1brl3*; **S11A and S11B** Fig) in the dark were not the ones with the strongest deviation from the wild type under water deficit in the dark (**S8 Fig**). We also looked at the expression levels of the light responsive gene *LHCB1.2* via qPCR in wild-type Ws-2 versus *bin2-3bil1bil2*. The data show that *LHCB1.2* gene expression is light-regulated in *bin2-3bil1bil2* seedlings (**S12 Fig**). We further looked at the response of selected BR mutants to water stress (-0.4MPa) in the light. Here, we focused on the hypocotyl response as this was clear and consistent in the wild type (**S3A Fig**). We found that both *bin2-3bil1bil2* and *bzr1-1D* mutants were unimpaired in their hypocotyl responses to water stress in the light (**S13 Fig**). We conclude that the investigated BR mutants are not primarily impaired in their ability to perceive light or water stress.

## BIN2 is required for a differential regulation of cell anisotropy in the hypocotyl

Hypocotyl elongation in the dark is known to occur via cellular elongation, with no significant contribution of cell division in the epidermis or cortex [39]. An assessment of organ and cell length in this study corroborated these findings, also for water stress in the dark. Indeed, we show that cellular parameters (cell length and width, assessed in scanning electron micrographs of hypocotyl cells) accounted for more than 73% of the observed differences in organ length and >89% of the differences in organ width (see **Fig 5A and 5B** for cellular parameters; **S5A and S5B Fig** for organ length and width and **Fig 5G and 5H** for a computation of fold-changes). We, therefore, conclude that it is predominantly a cellular response that controls hypocotyl growth under our conflict-of-interest scenario. To explore whether general BR-related cell elongation defects led to the confused phenotypes of some BR pathway mutants, we analysed *bin2-1* mutants, which were among the most severely impaired hypocotyl response to water stress in the dark (**S8A Fig**). The data show a most striking impact of *bin2-1* on growth anisotropy, assessed in 2D as length/width (**Fig 5F**). Indeed, in a comparison between dark and dark with water stress (darkW), the anisotropy of hypocotyl cells decreased considerably in the wild type (**Fig 5C**), but showed no adjustment in *bin2-1* (**Fig 5F**). Cell length alone showed the elongation defect typical of *bin2-1* mutants, with a much greater deviation from the wild type under darkW than under dark or light conditions; nonetheless, there was a significant length adjustment to water stress in the dark, even in *bin2-1* (**Fig 5E**). These observations suggest that the impaired *bin2-1* hypocotyl response can be attributed to an inability to differentially regulate cell anisotropy in response to the simultaneous withdrawal of light and water.

## The BIN2 clade of shaggy-like kinases differentially regulate the timing and extent of cell elongation in the root apical meristem in response to additive stress

It is generally accepted that root growth correlates with the size of the root apical meristem (RAM; [40]). Meristem size was assessed by computing the number of isodiametric and transition cells ([41,42]; **S8 Method**). In addition, we applied a Gaussian mixed model of cell length to distinguish between short meristematic cells and longer cells in the elongation zone (**S14 Fig** and **S8 Method**; [43]). Meristem size was shortest under water deficit in the dark (**Figs 6A**,

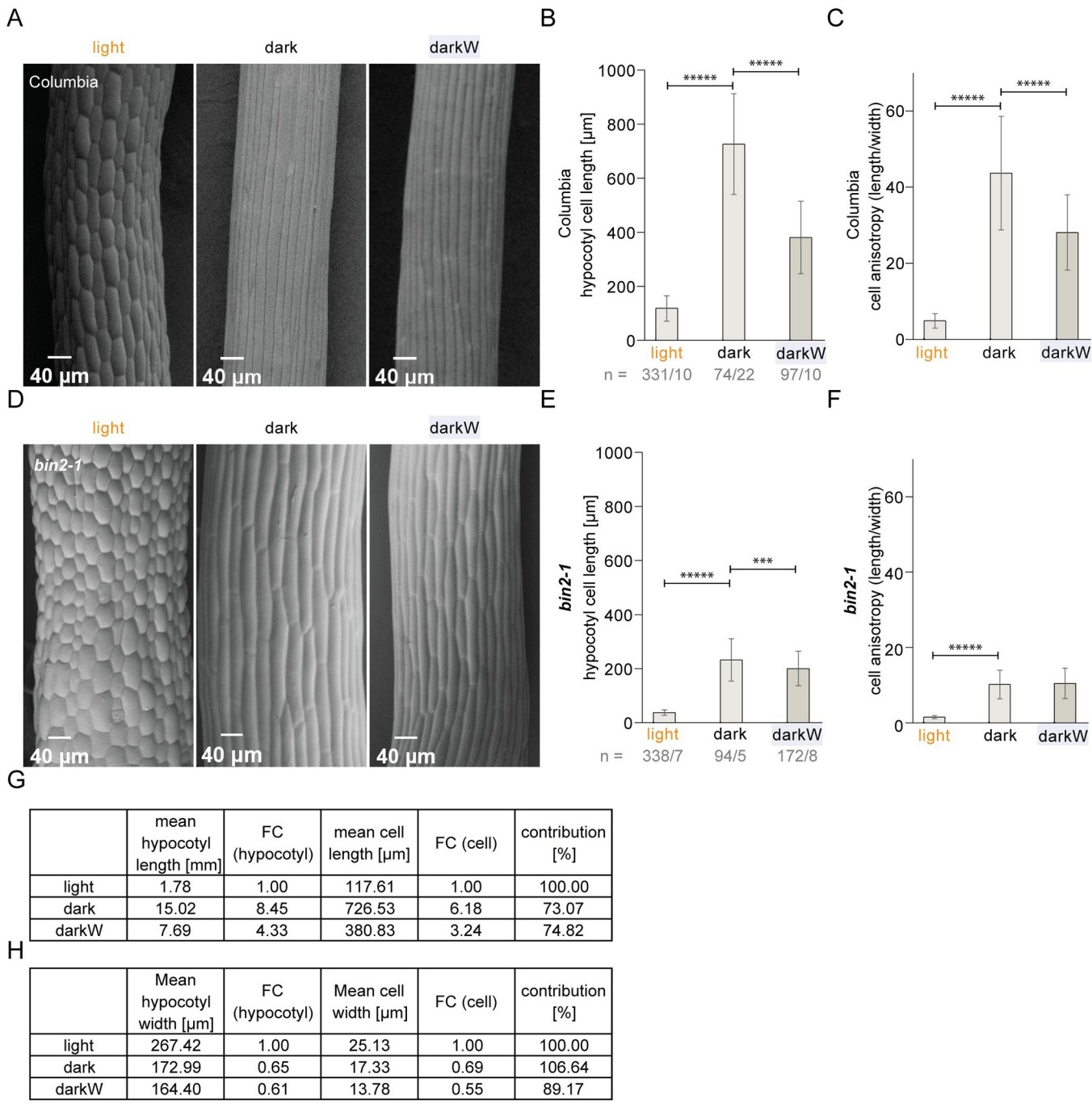

**Fig 5. Hypocotyl properties under single versus multiple stress conditions in wildtype and *bin2-1* mutants.** Seed were germinated on 1/2MS medium and incubated for ten days under light, dark or darkW conditions. (A-C) Col-0 wild type; (D-F) *bin2-1* semi-dominant allele. Scanning electron micrographs (A, D) were used to assess cellular parameters. (B, E) hypocotyl cell length. (C, F) hypocotyl cell anisotropy measured in 2D as length/width. Notice the highly significant decrease in anisotropy in the wild type (C) but the lack of response in *bin2-1* (F) for darkW as compared to dark. (G, H) A comparison of organ versus cell parameters for the Col-0 wild type under the different environmental conditions, with a computation of fold-changes (FC) with respect to the light condition. Mean cell width significantly (P < 0.00001; not shown) decreased between light and dark and again between dark and darkW (H). Note that, as depicted in the last columns labelled "contributions", the cellular parameters can account for 73–74% of organ length and 89–106% of organ width. The sample size (n) is given as the number of cells/ number of seedlings that were analysed. P-values were computed with a two-tailed student's T-test and are represented as follows: ***: 0.001–0.0001; *****: < 0.00001. See related **Fig 7**.

S15A, **and** S15B) and, surprisingly, did not correlate well with final organ length (**Figs** 1C **and** 6G). We, therefore, measured mature cell length. This was highest in the dark, the condition with the shortest roots (**Fig 6B**). Thus, neither meristem size nor mature cell length account for the fold-change in final organ length (**Fig 6G**). To address this counterintuitive result, we deployed the CycB1,1:GUS [44] marker, expressed exclusively in cells undergoing M phase (S15C **and** S15D **Fig**). We observed an eight-fold higher number of mitotic cells in darkW as compared to dark conditions (S15C **Fig**). This difference was enhanced at an earlier time point (**Figs** 6C **and** S15C) and could explain the longer organ length but not the smaller meristem size of roots under darkW versus dark (**Fig 6G**).

We subsequently looked at cell length, width and anisotropy (computed in 2D as length/ width) along single epidermal cell files as a function of distance from the quiescent centre. Cell elongation occurred in cells closer to the QC under darkW (blue arrowhead in **Fig 6D**) as compared to dark or light conditions (black and orange arrowheads in **Fig 6D**). Furthermore, the slopes of both the length and anisotropy curves were steepest for darkW (green arrows in **Fig 6D and 6F**). This was also apparent when we looked at the length of the first elongated cell, which was highest in darkW (S15E **Fig**). Cell elongation in cells close to the QC translates into an early exit from the root meristem. We conclude that root growth under water deficit in the dark is due not only to increased cell division but also to an early exit from the meristem (**Fig 7B**).

We then investigated root meristem properties in *bin2-3bil1bil2*, which had the most aberrant root response to water stress in the dark (**Figs** 3E **and** S8B magenta arrow). Meristem size and mature cell length followed the same trends in a comparison between *bin2-3bil1bil2* (S16A **and** S16B **Fig**) and the wild type (**Fig 6A and 6B**), but the extent of elongation in cells proximal to the QC differed (S16C **Fig**). Indeed, *bin2-3bil1bil2* length and anisotropy curves lacked the steep slopes characteristic for darkW in the wild type (compare the green arrows in **Fig 6D, 6F and 6J** to the purple arrows in **Figs** 6J **and** S16C). We conclude that *bin2-3bil1bil2* mutants fail to adjust their root length due to an inability to differentially regulate the elongation of meristematic cells in the root in response to water stress in the dark.

Our observations suggest that root growth under our conflict-of-interest scenario requires root apical meristem function as well as the differential regulation of cell length in the elongation zone. To address this hypothesis, we turned to PLETHORA (PLT) AP2-domain transcription factors, which play a pivotal role in the read-out of the auxin gradient in root tips [45]. Indeed, high levels of PLT activity at the stem cell niche promote stem cell identity and maintenance; intermediate levels at the transition zone promote mitotic activity; and low levels in the elongation zone are required for cell differentiation [45]. Interestingly, *plt1plt2* mutants [46] had an unimpaired hypocotyl response but failed to elongate their roots in response to water stress in the dark (S15F **Fig**). Taken together, the cell length and anisotropy curves (**Fig 6**) and genetic analyses (**Figs** 6, S15F **and** S16) suggest that root length under our different environmental conditions is regulated by (i) the mitotic index, (ii) the timing and extent of cell elongation—translating into the timing of meristematic exit—and (iii) cell geometry. We also conclude that these are differentially modulated to account for increased root length under different environmental conditions (**Fig 6C–6E**). In addition, an analysis of root meristem properties in *bin2-3bil1bil2* suggests that the BIN2 clade of shaggy-like kinases is required for the Arabidopsis seedling's ability to deploy different root growth strategies in response to abiotic stress cues under limiting conditions.

## Discussion

This study establishes screen conditions that enable us to monitor trade-offs between hypocotyl versus root growth in the Arabidopsis seedling. Our screen is based on a conflict-of-

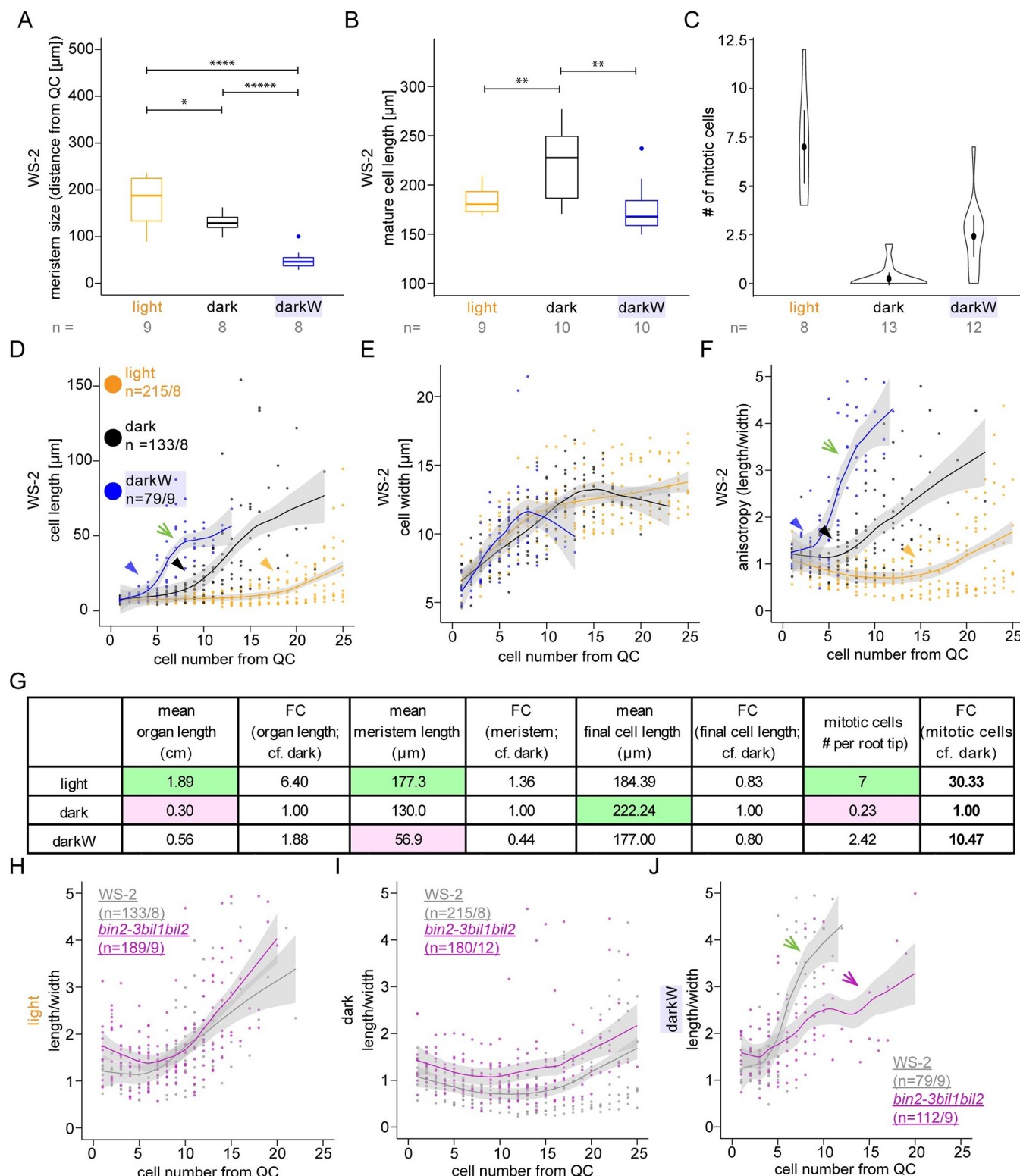

**Fig 6. Root meristem properties under single versus multiple stress conditions.** Seedlings were grown in the light (orange), dark (black) or dark with -0.4MPa water stress (blue). (A-F) Wild type, with ecotype specified in the panel. (A) Meristem size determined via mixed Gaussian models, as described ([43]; **S14 Fig**). (B) Mature cell length, based on the ten most elongated cells for each condition; similar conclusions were reached when the 50 longest cells were used. (C) mitotic index at day 7 based on CycB1,1:GUS ([44]; **S15C and S15D Fig**). (D-F and H-J) 10 days after incubation, single epidermal cell files were measured, starting at the epidermal/ lateral root cap initials. The fitted lines were generated with Local Polynomial Regression Fitting with the 'loess' method in R; grey

shading designates the 95 percent confidence interval. (D-F) cell lengths (D), width (E) and anisotropy (in 2D as length/width; F) of consecutive cells as a function of cell number from the quiescent centre (QC); the green arrows point to the steep slope for length (D) and anisotropy (F) under the darkW condition and the arrowheads to the kinks in the curves–the initiation of elongation—under all three conditions. (G) A tabulation of fold-changes (FCs) of measured parameters between different environmental conditions in the wild type (Ws-2, Col-0), with the smallest number highlighted in pink and the largest in green; note that the only FCs that go in same direction as root organ length are for the mitotic index (bold; data depicted in panel C). (H-J) A direct comparison of cell anisotropy under different environmental conditions between the *bin2-3bil1bil2* triple mutant (purple) and the corresponding Ws-2 wild type (grey); notice that the mutant most markedly deviates from the wild type (compare purple versus green arrows in J) in the darkW condition (J), where the steep slope characteristic of the wild type is replaced by a flatter, undulating curve. The sample size (n) is given as the number of seedlings in panels a-c and as the number of cells/ number of seedlings that were analysed in panels H-J. P-values were computed with a two-tailed student's T-test and are represented as follows: *: 0.05–0.01; **: 0.01–0.001; ****: 0.0001–0.00001, *****: < 0.00001. See related **Figs 7, S14, S15,** and **S16**.

interest hypocotyl-versus-root scenario consisting of the simultaneous withdrawal of light and water. A question we addressed was how nutrients, light and water are prioritized in the germinating seedling. We observed a clear priority for hypocotyl growth in search of light over primary root growth in search of nutrients. In contrast to light and nutrients, light and water stress appeared to be equally prioritized in the germinating seedling. Shoot and root lengths were exquisitely fine-tuned to the wavelength and intensity of the light source, and to the severity of water stress. An assessment of organ and cell length suggested that hypocotyl elongation occurred predominantly via cellular elongation (**Fig 7A**). In contrast, root growth appeared to be regulated by a combination of cell division and the timing of exit from the meristem (**Fig 7B**). We have shown that the BR pathway is implicated in hypocotyl versus root trade-offs. Gain- or loss- of function *bin2* alleles were most severely impaired in their ability to adjust cell geometry in the hypocotyl or cell elongation as a function of distance from the quiescent centre in the root tips.

It is generally accepted that root length correlates with meristem size. Using kinematic methods, it has also been shown that accelerating root elongation is driven predominantly by an increased number of dividing cells [40]. Conversely, root growth cessation in response to salt stress has been shown to be a result of decreased cell division, correlating with a reduced meristem size, as well as a reduced mature cell length [47]. Therefore, it appears counterintuitive that meristem size and organ length do not correlate in our conflict-of-interest scenario. Questions arise as to why the meristem is smaller under water deficit in the dark even though the mitotic index is higher than in the dark, and how growth is promoted under our additive stress scenarios. An important difference between our conditions and those described by others is that we germinated seed under limiting conditions in the dark in the absence of a carbon source; related studies (such as [24]) were carried out in the light or added sucrose to the growth medium in the dark, such that seedlings were not limited with respect to available energy. In this study, we observed growth arrest in the dark, as seen by the low number of mitotic cells in root tips. When water stress was applied in the dark, the mitotic index increased, but the newly produced meristematic cells immediately elongated, thereby exiting the meristem. As a consequence, meristem size remained small despite the increased number of mitotic cells. It appears that what our study shows is a novel paradigm for root growth under limiting conditions, which depends not only on shoot-versus-root trade-offs in the allocation of limited resources, but also on an ability to deploy different strategies for growth in response to abiotic stress cues.

The simplest conceptual framework to explain our observations evokes not only differential regulation of hypocotyl versus root growth, but also hypocotyl to root and root to hypocotyl signalling (**Fig 7C**). As water stress was applied exclusively to the root but also impacted hypocotyl growth (**Figs 1E and S3D**), we evoke a root to hypocotyl (acropetal) signal to coordinate trade-offs in organ growth in response to water stress (**Fig 7C** blue arrow). Conversely, we postulate that a hypocotyl to root (basipetal) signal coordinates trade-offs in organ growth in

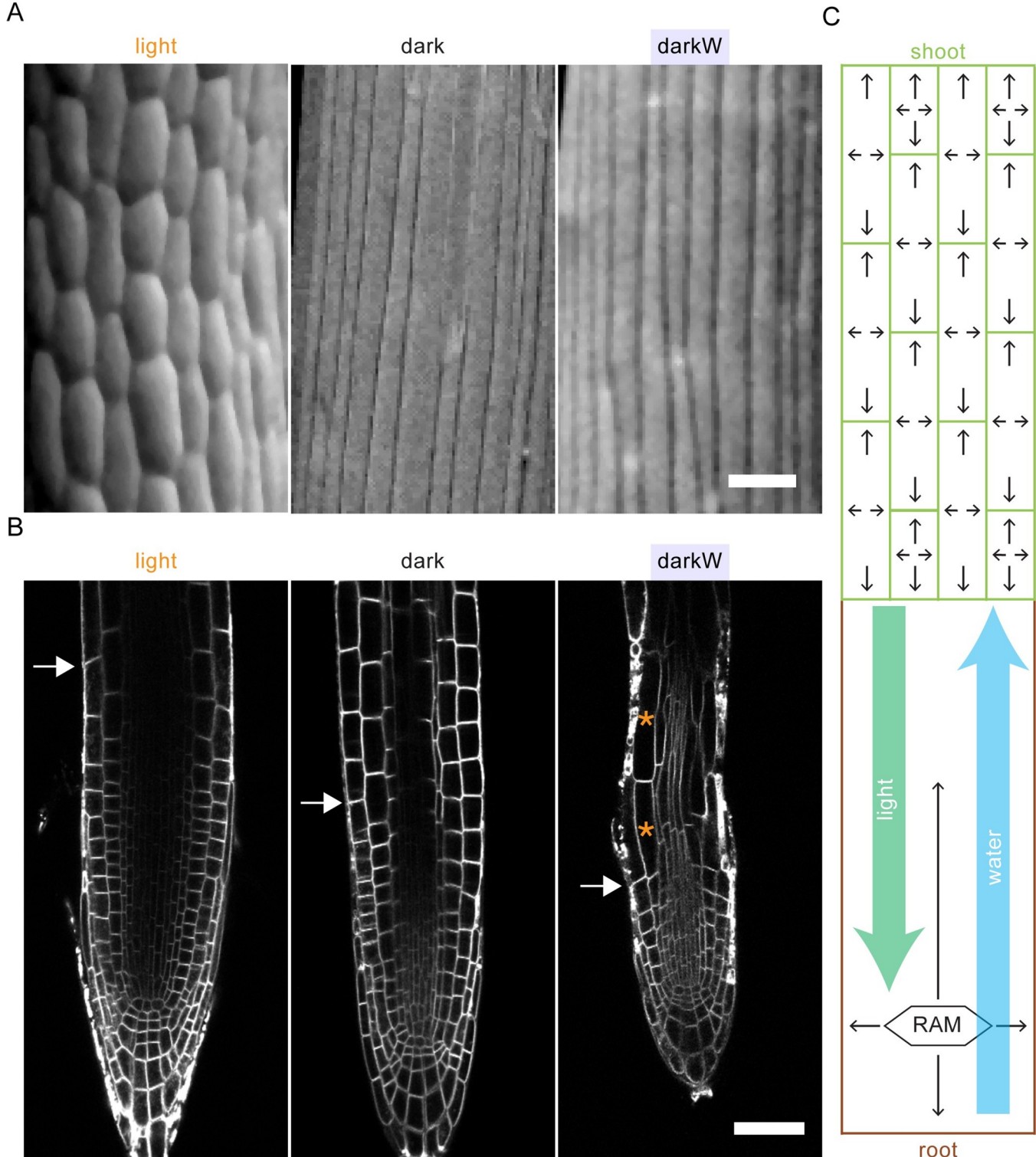

**Fig 7. Growth regulation and information flow in response to the availability of water and light.** (A) Scanning electron micrographs of wild-type (Col-0) hypocotyls under different environmental conditions. (B) Confocal micrographs of root tips under different environmental conditions; white arrows mark the junction between the meristematic and elongation zones (depicted as mean meristem size in **Fig 6A**) as assessed by mixed Gaussian model analysis ([43]: **S14 Fig**) and orange asterisks mark the highly elongated cells at the very beginning of the elongation zone under darkW. Wild type (Col-0) for light and dark; P_BRI1::BRI1-GFP for darkW; FM4-64 signal in all cases. Scalebar: 50 μm. (C) Information-flow diagram as a translation of our empirical findings. The green rectangles in the upper part of the diagram represent hypocotyl cells, and the arrows within each cell depict growth as a decentralized cellular response. The

lower part of the diagram (brown-beige) represents the root system, with growth responses (arrows) driven in part by the root apical meristem (RAM). The green arrow depicts a hypothetical hypocotyl to root (basipetal) signal that coordinates trade-offs in organ growth in response to light; this does not take into account the possibility that photoreceptors in the root also perceive and respond to light [49]. Conversely, the blue arrow depicts a hypothetical root to hypocotyl (acropetal) signal that coordinates trade-offs in organ growth in response to water stress.

response to light (**Fig 7C** green arrow). However, and even though photoreceptors are considerably more abundant in the hypocotyl than in the root [48], it needs to be borne in mind that photoreceptors in the root could be playing a role in root responses to light or to darkness [49]. The impact of light was seen under all different wavelengths, suggesting that the light signal acts downstream of all photoreceptors.

There is a fair amount of controversy in the literature regarding the role of BR in cell elongation versus cell division [36]. It has, on the one hand, been argued that BR's impact on cell division [41] is a secondary consequence of a primary defect in cell elongation, as cells need to achieve a certain size prior to the initiation of the cell cycle [36,50]. On the other hand, brassinosteroids have been shown to control QC identity and to impact formative divisions in the root apical meristem [33,41]. Furthermore, high concentrations of BL have been shown to promote cell elongation and an early exit from the root apical meristem [41,51]. More recent models for BR signalling in the root show or simulate the greatest impacts on cell anisotropy, growth rates and division plane orientation [43,52]. In this study, we addressed the role of BR signalling in cell elongation in the hypocotyl and root tip within the context of growth trade-offs and conflict-of-interest scenarios. While an analysis of cell anisotropy in 3D was not within the scope of this study, our 2D analysis clearly showed that BIN2 was required for an adequate regulation of cell anisotropy in the hypocotyl in response to the simultaneous withdrawal of light and water. In the root, *bin2* higher order loss of function mutants were most severely impacted in their ability to differentially adjust cell length as a function of distance from the QC in response to environmental cues.

The BR pathway intersects with light and ABA signalling at many levels. Light is known to inhibit BR biosynthesis, to stabilize or activate BIN2 and to impact BR-responsive transcriptional regulation [53–57]; reviewed by [36,58,59]. BIN2 modulates light responses by phosphorylating PIFs, transcriptional regulators inhibited by phytochromes, thereby targeting them for degradation [60,61]. In addition, the interaction between BIN2 and its substrates is regulated by photoreceptors or light signalling components [57,61]. The BR pathway has also been implicated in drought responses and ABA signalling [31,62]. BIN2 is a target of PP2C phosphatases in the abscisic acid pathway and mediates drought responses by targeting ABA signalling components and drought or desiccation responsive transcription factors [63–69]. Furthermore, BIN2 homologues have been implicated in root responses to osmotic stress [70]. In addition to its role in light and drought responses, the BR pathway has been implicated in the growth versus defence trade-off [6,9] and in regulating hypocotyl elongation in response to far-red light and salt stress [21]. Studies on responses to abiotic stress factors have typically addressed growth arrest or trade-offs between growth and acclimation [71]. Indeed, root growth is inhibited by, for example, phosphate deprivation or salt stress [47,72]. Recent efforts have addressed strategies for engineering drought resistant or tolerant plants that do not negatively impact growth [30,73]. In contrast to other studies, here we look at two abiotic stress factors that promote organ growth. Indeed, hypocotyl growth is promoted by darkness or low light and primary root growth by water deficit in this study.

In the judgement and decision-making model for plant behaviour put forth by Karban and Orrock (2018) [2], signal integration might be considered integral to judgement. In decision theory one could refer to signal integration as an assessment of the "state of the world",

although in our case this state is integrated with endogenous developmental signals as well. In our conflict-of-interest scenario, the "confused" phenotypes of *bin2* mutants are indicative of an integral role in decision-making *per se*. Because *bzr1-1D* and *bes1-1D* gain of function mutants had relatively weak phenotypes while the *bri1-116 bzr1-1D* were severely impaired in our conflict-of-interest scenario, it is tempting to speculate that the strong phenotypes of *bin2* alleles are related not specifically to the role of BIN2 in BR signal transduction, but rather to its role as a key node in hormone crosstalk and signal integration. Whether judgement and decision making can be distinguished from each other empirically remains unclear. As the BIN2 clade of shaggy-like kinases regulates cell anisotropy and the timing and extent of cell elongation in the hypocotyl and root apical meristem, it may play a role not only in signal integration but also in the execution of decisions (or in an implementation of the action; [41,74]). Thus, this study does not enable us to empirically distinguish between decision making on the one hand and signalling and execution on the other.

Future experiments will address where in the seedling decision-making processes occur, the nature, interdependence and movement of the acro- and basipetal signals, and how these facilitate shoot to root communication to fine tune trade-offs between root versus shoot growth. 3D imaging will be required to assess the impact of abiotic stress and/or of BR signalling on different cell files or tissue layers in the root (see [43,52,75,76]). Similarly, time-lapse imaging will be required for temporal resolution. As a limited budget is an essential component of our screen conditions, the role of energy sensing and signalling [77] in growth trade-offs will need to be elucidated. In addition, phosphoproteomics may enable us to better understand BIN2 targets under our conflict-of-interest-scenario. The screen has broader implications for plant responses to multiple stress parameters applied simultaneously. Indeed, with changing climate and mounting degrees of uncertainty, a possibly comforting outcome of our screen is that although we actively looked for mutants with "confused" phenotypes, we were, with the exception of selected BR signalling mutants, hard put to uncover any. Rather, our overall conclusion is that even mutants with very severe growth defects were, when faced with extreme multiple stress conditions, by and large capable of adjusting their shoot to root growth trade-offs to optimize their chances of survival upon germination.

## Supporting information

**S1 Fig. The impact of nutrient stress (-K, -N, -P) on hypocotyl versus root growth in dark-grown seedlings.** Col-0 seed were plated on NPK media with or without (-) K, N or P and incubated for seven days in the dark. (A-D) germination in the absence of a carbon source. Panels (A-D) are from the same experiment. Minor fluctuations were detected, but no clear trade-offs between hypocotyl and root growth were observed when nutrient stress was applied to dark-grown seedlings. Based on the long hypocotyls (A) and short roots (B), there is a clear priority for light (translating into hypocotyl growth) over nutrients (translating into root growth) in the germinating seedling as seen in the hypocotyl/root ratio (C) and the total length (D). (E-H) germination in the presence (+) of 1% sucrose as a carbon source. The presence of sucrose as a carbon source increased the total length of the seedlings up to two-fold (h); numbers above the columns are fold-changes compared to the same condition without sucrose. (I, J) Panels in (I, J) depict the impact of a carbon source. In the presence (+) of a carbon source, hypocotyl length remained fairly constant but the roots were able to grow longer (up to 6-fold increase in root length, as compared to the absence (-) of sucrose). Thus, there was no clear trade-off, by which we refer to the growth of one organ at the expense of another. As the genetic screen was designed to mimic limiting conditions, we omitted sucrose from the media in all further experiments. The number (n) of seedlings measured per condition is in grey

below the graph. *P*-values were computed with a two-tailed student's *T*-test and are represented as follows: **: 0.01–0.001; *****: < 0.00001. Related to **Fig 1**.
(TIF)

**S2 Fig. Hypocotyl versus root growth in response to osmotic stress and salt stress in dark-grown seedlings.** Hypocotyl (A) and root (B) lengths of seedlings (Ler wild type) germinated under osmotic (100mM mannitol) or salt stress (50-100mM NaCl) in the dark in the absence of a carbon source. No reproducible differences in the ratio (C) were observed (P > 0.5). The seedling total length (D) decreased with increasing salt concentration. The number (n) of seedlings measured per condition is in grey below the graph. P-values were computed with a two-tailed student's T-test and are represented as follows: *: 0.05–0.01; **: 0.01–0.001; ***: 0.001–0.0001; ****: 0.0001–0.00001, *****: < 0.00001. Related to **Fig 1**.
(TIF)

**S3 Fig. Trade-offs between hypocotyl and root growth in response to water stress.** (A) Hypocotyl/root ratio of seedlings (Col wild type) germinated in the light with (lightW) or without (light) water stress (PEG -0.4MPa); the clearest response to water stress in the light is a reduction in hypocotyl length (left panel). This is consistent with the observations of van der Weele et al. [24] on the use of PEG in the light. (B) Seed (Ler) were geminated in the dark on MS medium (- 0.2 MPa) with a gradient of water stress ranging from -0.2 to -0.8 MPa. The hypocotyl/root ratio of Ler seedlings is similar to that of Col-0 (**Fig 1D**): water stress applied in the dark increases root length at the expense of hypocotyl length, giving rise to a decrease in the hypocotyl/root ratio. (C) The length of wild type (Col-0) seedlings was seen to decrease with increasing water stress after seven (but not after ten days; cf. **Fig 1E**), and this decrease was observed in control experiments to be due to delayed germination induced by water stress. (D) Roots were stained with methanol blue; arrows point to the junction between the hypocotyl and root and arrowheads to the end of the root. The number (n) of seedlings measured per condition is in grey below the graph. P-values were computed with a two-tailed student's T-test and are represented as follows: *: 0.05–0.01; **: 0.01–0.001; ***: 0.001–0.0001; ****: 0.0001–0.00001, *****: < 0.00001. Related to **Fig 1** and **S5 Fig**.
(TIF)

**S4 Fig. Impact of different light conditions and light signalling on hypocotyl versus root growth.** (A-C) Wild-type (Col-0) seedlings were germinated on MS media under different light conditions, with or without water stress. (A) Blue light at varying intensities ranging from 5.3 to 0 μmol m$^{-2}$ s$^{-1}$. (B) Red light at varying intensities ranging from 4.2 to 0 μmol m-2 s-1. (C) Far-red light at 0.5 μmol m-2 s-1, compared to dark grown seedlings. A decreasing intensity gradient of red light, as well as low levels of far-red light, increases hypocotyl length at the expense of root length as seen in the hypocotyl/root ratio. (D-G) Quadruple *cry1 cry2 phyA phyB* mutant seedlings (F, G) had no hypocotyl response to light (F; white light intensity: 250 μmol m$^{-2}$ s$^{-1}$, compared to dark conditions) but had highly significant responses to water stress applied in the dark (G); corresponding Ler wild type (D, E). The number (n) of seedlings measured per condition is in grey below the graph. Benjamini–Hochberg corrected P-values are represented as follows: *: 0.05–0.01; **: 0.01–0.001; ***: 0.001–0.0001; ****: 0.0001–0.00001, *****: < 0.00001. Related to **Fig 1**.
(TIF)

**S5 Fig. Impact of water stress on hypocotyl length, width and volume in dark-grown seedlings.** Wild-type seeds (Col-0) were germinated on ½ MS in the dark with or without water stress. (A) Hypocotyl and root lengths, as well as hypocotyl/root ratio. Note that seedlings germinated under water stress in the dark have shorter hypocotyls and longer roots than in the

dark, giving rise to a decreased hypocotyl/root ratio. (B) Hypocotyl width; hypocotyls are wider in light than in the dark; water stress in the dark results in a decrease in hypocotyl width. (C) Hypocotyl volume; the volume increases in the dark (cf. light) and decreases in response to water stress in the dark. The number (n) of seedlings measured per condition is in grey below mean ±StDev bar graphs. *P*-values were computed with a two-tailed student's *T*-test and are represented as follows: **: 0.01–0.001; ***: 0.001–0.0001; ****: 0.0001–0.00001, *****: < 0.00001. Scale bars = 1mm. See related **Figs 1** and **S3**.
(TIF)

**S6 Fig. Light responses in the B1 mutant and comparison to *bin2-1*.** (A) Light responses were attenuated in B1, with a shorter hypocotyl and longer root in the dark. (B) RQ$_{ratios}$ (see main methods) were comparable and attenuated in both B1 and *bin2-1*. (C) The total length is depicted as the hypocotyl (top, green) and root (bottom, beige). B1 was shorter than Ler (L) in the light and in the dark, but not under darkW conditions. The number (n) of seedlings measured per condition is in grey below the graph. P-values were computed with a two-tailed student's T-test and are represented as follows: *: 0.05–0.01; **: 0.01–0.001; ***: 0.001–0.0001; ****: 0.0001–0.00001, *****: < 0.00001. Related to **Fig 2**.
(TIF)

**S7 Fig. The response of BR pathway mutants to water stress in the dark: violin plots.** Violin plots of the hypocotyl, root and ratio responses of BR pathway mutants, with the corresponding wild-type ecotype as reference. The dot represents the mean and the line the 95% confidence interval. Note the high variance of the wild-type (Col-0) root response under darkW. Mutant alleles are described in **S1 Table**; null alleles are depicted in regular font, semi-dominant or dominant in bold and higher order mutants are underlined. Datasets are as in **Fig 3**, with the exception of *bzr1-1D* and *bri1-116 bzr1-1D*, where different representative replicates are shown.
(TIF)

**S8 Fig. The response of BR pathway mutants to water stress in the dark.** Response quotients (RQ, left) and volcano plots (right) of the hypocotyl and root responses water stress in the dark. RQs are normalized to the wild-type ratio quotient; a value of 1 (vertical red line) indicates that the response to a shift from dark to darkW is similar to that of the respective wild-type ecotype. Each replicate is represented by a dot; purple dots are for initial and grey dots for optimized screen conditions; blue dots are for data from SEM measurements. Volcano plots with the mean RQ depicted on the left on the *X*-axis and the *P*-Value of the response on the *Y*-axis (negative log scale; a median of all replicates was used). Null alleles are depicted in regular font, semi-dominant or dominant in bold and higher order mutants are underlined. (A) Hypocotyl responses to dark versus darkW conditions. Note that *bri1-116 bzr1-1D* and the semi-dominant *bin2-1* have an attenuated hypocotyl response, RQ$_{hypocotyl}$ (magenta arrows). This was not observed in the triple *bin2-3bil1bil2* knock out (green arrow). (B) Root responses to dark versus darkW conditions. Note that the triple *bin2bil1bil2* knock out has the strongest RQ$_{root}$ phenotype (magenta arrow). This is in contrast to *bin2-1* (green arrow). Thresholds used to interpret the results are tabulated at the bottom of the figure; magenta colour indicates an attenuated and green an exaggerated response. Related to **Figs 3** and **4**.
(TIF)

**S9 Fig. Seed weights in selected BR mutants.** The number (n) of seed bags analysed per condition is in grey below mean ±StDev bar graphs. Each seed-bag contained on average 397 seed. *P*-values were computed with a two-tailed student's *T*-test and were non-significant: >0.05.

Related to **Figs 5 and 6**.
(TIF)

**S10 Fig. Light responses in BR pathway mutants: bar graphs.** Seedlings were germinated on ½ MS in the light and dark (A) Col-0 (wild type). (B) BR biosynthesis mutant *cpd*. (C) BR perception mutant *bri1brl1brl3* (D) BR signalling mutant *bin2-1* (a semidominant gain of function allele). (E) *bin2-3bil1bil2* triple knockout; (F) Transcription factor mutant *bzr1-1D*, a dominant allele. (G) Bypass mutant *bri-116 bzr1-1D*. (A-M) Null alleles are depicted in regular font, semi-dominant or dominant in bold and higher order mutants are underlined. Note that all mutants had significant responses to light versus dark conditions. At least 3 experiments were performed for each line, and a representative one is shown here on the basis of RQ and P values (see **S11 Fig**). The number (n) of seedlings measured per condition is in grey below the mean ±StDev bar graphs. P-values were computed with a two-tailed student's T-test and are represented as follows: *: 0.05–0.01; **: 0.01–0.001; ***: 0.001–0.0001; ****: 0.0001–0.00001, *****: < 0.00001. For mean RQ values and median P-values see **S11 Fig**. Ecotypes are described in **S1 Table**. Related to **Figs 3 and 4**.
(TIF)

**S11 Fig. Light responses in BR pathway mutants.** Response quotients (RQ, left) and volcano plots (right) of the hypocotyl, root or ratio responses to light versus dark. RQs are normalized to the wild-type ratio quotient; a value of 1 (vertical red line) indicates that the response to a shift from light to dark is similar to that of the respective wild-type ecotype. Each replicate is represented by a dot; grey dots are for optimized screen conditions; blue dots are for data from SEM measurements. Volcano plots with the mean RQ depicted on the left on the X-axis and the P-value of the response on the Y-axis (negative log scale; a median of all replicates was used). Null alleles are depicted in regular font, semi-dominant or dominant in bold and higher order mutants are underlined. (A) Hypocotyl responses to light versus dark conditions. Note that *cpd*, *bri1brl1brl3* and *bin2-1* mutants have a severely attenuated hypocotyl response $RQ_{hypocotyl}$. (B) Root responses to light versus dark conditions. Note that *cpd*, and *bin2-1* mutants have the strongest $RQ_{root}$ phenotype, and that *bin2-1* gain of function and *bin2-3bil1bil2* loss of function mutants have opposite root phenotypes. (C) Hypocotyl/root ratio responses to light versus dark conditions. In all three volcano plots, *cpd* and *bin2-1* mutants are the most severely impaired (most attenuated response (RQ), lowest P-value). Thresholds used to interpret the results are tabulated at the bottom of the figure; magenta colour indicates an attenuated and green an exaggerated response. Related to **Figs 3 and 4**.
(TIF)

**S12 Fig. Expression of light responsive gene *LHCB1.2* in Ws-2 wild type (WT) and *bin2-3bil1bil2* mutant seedlings.** Seed were germinated on 1/2 MS plates and incubated in the light (orange; +) or dark (black; -) for 10 days. Transcript abundance was determined by qRT-PCR with *UBIQUITIN-PROTEIN LIGASE-LIKE PROTEIN* as a reference for normalization (see **S7 Method**). Expression in the light was set at 100% for both genotypes. Gene expression was significantly downregulated in the dark, in both the wild type and in *bin2-3bil1bil*2 mutants. Data represent means ± StDEV of at least three independent experiments, where each measurement was based on three technical replicates. Three independent lines were used for the mutant. *P*-values were computed with a two-tailed student's *T*-test and are represented as follows: **: 0.01–0.001; *****: < 0.00001.
(TIF)

**S13 Fig. Responses of selected BR mutants to water stress in the light (lightW).** Hypocotyl responses of (A) *bin2-3bil1bil2* triple knockout; (B) Transcription factor mutant *bzr1-1D*, a

dominant allele. The number (n) of seedlings measured per condition is in grey below the graph. *P*-values were computed with a two-tailed student's *T*-test and are represented as follows: *****: < 0.00001 (C) RQ$_{hypocotyl}$ response quotient of the hypocotyl under light/lightW conditions, normalized to the wild-type response quotient; a value of 1 (vertical red line) indicates that the response to a shift from light to lightW is similar to that of the respective wild-type ecotype. Each replicate is represented by a dot. (D) Volcano plot with the mean RQ$_{hypocotyl}$ depicted in (C) on the *X*-axis and the median *P*-Value of the response on the *Y*-axis (negative log scale; a median of all replicates was used). Related to **Figs 3 and 4**.
(TIF)

**S14 Fig. Root apical meristem size under different environmental conditions.** As described by Fridman et al. [43] we used the expectation maximization algorithm as implemented in the mixtools R package to fit a two-Gaussian mixture model to the cell length parameter in each condition. This captures two populations of short versus long cells. Cells with a probability > 0.8 of being in the short-length Gaussian were considered to be meristematic cells (blue). Cells outside the meristem were considered as being elongating cells (red). (A-C) wild type (Ws-2). (D-F) *bin2-3bil1bil2* triple null BR signalling mutant. See **S8 Method**.
(TIF)

**S15 Fig. Root apical meristem properties under different environmental conditions.** (A) A shift from light to darkness decreases the number of isodiametric cells (defined as being not longer than wide; $P = 1.2E^{-08}$) and a concomitant decrease in meristem size. (B) In the dark, water stress results in a decrease in the number of transition cells (includes the first cell being longer than wide up to the last cell whose length is <150% that of the previous cell; see Materials and Methods). (C; D) Seedlings expressing the M-phase CycB:GUS marker and stained with X-Gal at day 7 (D) and at days 7 versus 10 (C). The number of cells expressing CycB:GUS cells per root tip decreased from light to dark (D) but increased from dark to darkW (C, D) at both days 7 and 10. (E) The The first elongated cell was longer under darkW (see orange asterisk in Fig 7B) than under dark conditions. (F) *plt1plt2* mutants have an unimpaired hypocotyl response but fail to reproducibly elongate their roots in response to water stress in the dark ($P_{root} = 0.27$). The number (n) of seedlings measured per condition is in grey below the mean ±StDev bar graphs. *P*-values were computed with a non-parametric Mann-Whitney-U tests with a Benjamini–Hochberg correction in (C) and with a two-tailed student's *T*-test in panels A, B, E, F; they are represented as follows: *: 0.05–0.01; **: 0.01–0.001; ****: 0.0001–0.00001; *****: < 0.00001. Scale bars: 50 µm. Related to **Fig 6**.
(TIF)

**S16 Fig. Root meristem properties in *bin2-3bil1bil2* under multiple stress conditions.** *bin2-3bil1bil2* seed were germinated in the light (orange), dark (black) or dark with -0.4MPa water stress (blue). 10 days after incubation, single epidermal cell files were measured, starting at the epidermal/ lateral root cap initials. (A) Meristem size determined via mixed Gaussian models, as described ([43]; see **S14 Fig**). (B) Mature cell length, based on the ten most elongated cells for each condition. (c, d) Cell lengths (C) and width (D) of consecutive cells as a function of cell number from the quiescent centre (QC); the fitted lines were generated with Local Polynomial Regression Fitting with the 'loess' method in R and grey shading designates the 95 percent confidence interval. The purple arrows point to the relatively flat slope for cell length (cf. green arrows pointing to the steep slope characteristic of the wild type in **Fig 6D**) in the darkW condition (C). The sample size (n) is given as the number of seedlings in panels a and b and as the number of cells/ number of seedlings that were analysed in panels C, D. P-values were computed with a two-tailed student's T-test and are represented as follows: *: 0.05–0.01; ***: 0.001–

0.0001; *****: $< 0.00001$. Related to **Fig 6**.
(TIF)

**S1 Table. Lines used in this study.**
(PDF)

**S2 Table. Segregation analysis of the B1 mapping population.**
(PDF)

**S1 Data. The numerical data for the main.**
(XLSX)

**S2 Data. The numerical data for the supporting information.**
(XLSX)

**S1 Method. Composition of nutrient stress plates.**
(PDF)

**S2 Method. Supplemental information on light experiments.**
(PDF)

**S3 Method. Preparation of PEG plates.**
(PDF)

**S4 Method. Seed handling for screen.**
(PDF)

**S5 Method. Picking, scanning, phenotyping and genotyping.**
(PDF)

**S6 Method. Positional Cloning.**
(PDF)

**S7 Method. Real-time PCR analysis.**
(PDF)

**S8 Method. Root apical meristem properties.**
(PDF)

## Acknowledgments

We thank Prof. Grill for his support. Alexander Christmann, Mikhal Kepka and other members of the Botany department provided EMS-mutagenized seed and made useful suggestions. Caroline Klaus tended to our plants in the green house and made valuable phenotypical observations. We are exceedingly grateful to Andreas Czempiel and Knut Thiele for technical assistance, and to Yannik Schreckenberg for curating the numerical data. Fiona Pachl initiated the positional cloning of B1. We thank Leon Assaad for insights into decision theory. Laura Gräbener, Manuel Jeller and Tobias Weiser contributed to this study as undergraduate students. Urs Schmidhalter provided expertise for nutrient stress. We thank Frej Tulin and Katarzyna Retzer for a critical appraisal of the manuscript and Joe Kieber for useful discussion. The NASC stock centre distributed public seed stocks and Zhiyong Wang, Joanne Chory, Jorge José Casal and John Celenza shared published resources. We thank Roman Meier at the TUM-mesa facility, directed by Leonardo Teixera, for supporting us with optimal growth conditions for our plants. We gratefully acknowledge the WZW/TUM Centre for Advanced Light Microscopy (CALM), headed by Ramon Torres-Ruiz, for unlimited access to confocal microscopes.

## Author Contributions

**Conceptualization:** Nils Kalbfuß, Alexander Strohmayr, Wojciech Palubicki, Cordelia Bolle, Farhah F. Assaad.

**Data curation:** Nils Kalbfuß, Alexander Strohmayr, Marcel Kegel, Lien Le, Friederike Grosse-Holz, Katharina Stöckl, Christian Wiese, Sophia Prem, Shuyao Sha, Katrin Franz-Oberdorf, Eva Facher, Cordelia Bolle, Farhah F. Assaad.

**Formal analysis:** Nils Kalbfuß, Alexander Strohmayr, Marcel Kegel, Lien Le, Friederike Grosse-Holz, Barbara Brunschweiger, Katharina Stöckl, Christian Wiese, Carina Franke, Caroline Schiestl, Sophia Prem, Shuyao Sha, Katrin Franz-Oberdorf, Marc Thiemé, Eva Facher, Cordelia Bolle, Farhah F. Assaad.

**Funding acquisition:** Cordelia Bolle, Farhah F. Assaad.

**Investigation:** Nils Kalbfuß, Alexander Strohmayr, Marcel Kegel, Lien Le, Friederike Grosse-Holz, Barbara Brunschweiger, Katharina Stöckl, Christian Wiese, Carina Franke, Caroline Schiestl, Shuyao Sha, Katrin Franz-Oberdorf, Marc Thiemé, Eva Facher, Cordelia Bolle, Farhah F. Assaad.

**Methodology:** Nils Kalbfuß, Alexander Strohmayr, Marcel Kegel, Lien Le, Friederike Grosse-Holz, Barbara Brunschweiger, Carina Franke, Katrin Franz-Oberdorf, Cordelia Bolle, Farhah F. Assaad.

**Project administration:** Farhah F. Assaad.

**Resources:** Eva Facher, Farhah F. Assaad.

**Supervision:** Nils Kalbfuß, Christian Wiese, Eva Facher, Wojciech Palubicki, Cordelia Bolle, Farhah F. Assaad.

**Validation:** Nils Kalbfuß, Marcel Kegel, Lien Le, Friederike Grosse-Holz, Cordelia Bolle, Farhah F. Assaad.

**Visualization:** Nils Kalbfuß, Alexander Strohmayr, Barbara Brunschweiger, Sophia Prem, Wojciech Palubicki, Cordelia Bolle, Farhah F. Assaad.

**Writing – original draft:** Farhah F. Assaad.

**Writing – review & editing:** Nils Kalbfuß, Alexander Strohmayr, Marcel Kegel, Lien Le, Christian Wiese, Juliane Hafermann, Wojciech Palubicki, Cordelia Bolle, Farhah F. Assaad.

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
