## [Decision Letter · Decision Letter 0]

12 Oct 2022

Dear Dr Assaad,

Thank you very much for submitting your Research Article entitled 'A role for brassinosteroid signaling in decision-making processes in the Arabidopsis seedling.' to PLOS Genetics.

The manuscript was fully evaluated at the editorial level and by independent peer reviewers. The reviewers appreciated the attention to an important topic but identified some concerns that we ask you address in a revised manuscript.

We therefore ask you to modify the manuscript according to the review recommendations. Your revisions should address the specific points made by each reviewer.

Yours sincerely,

Claudia Köhler

Section Editor

PLOS Genetics

Reviewer's Responses to Questions

**Comments to the Authors:**

Reviewer #1: The study is timely and important to understand how plants cope with manifold stimuli to adjust growth by orchestrating cell elongation rate. All my concerns were addressed, and I want to endorse the study for publication.

Reviewer #2: This manuscript by Kalbfuß et al identifies a role for brassinosteroid signaling in a decision-making process of root vs shoot growth under the combination of darkness and water stress. We reviewed a previous version of this manuscript and the authors have now adequately addressed many of our concerns. They provided new data in the form of BR mutant analysis and interesting observations showing that the increase in root length in response to water deficit in the dark appears to operate through increased cell divisions despite the smaller meristem size. Overall the revised manuscript is solid and well written. We have several relatively minor comments that should be addressed which are listed below.

1. Given the lot-to-lot and plate-to-plate variability that the authors note with PEG assays, can they be confident in claiming that variable hypocotyl versus root lengths are a read-out for a “confused phenotype”? It’s not clear that this claim is essential in their story demonstrating the root-versus-shoot tradeoffs mediated by BRs.

2. Both gain-of-function (bin2-1) and loss-of-function alleles (bin2-3bil1bil2) of BIN2 seem to be affected by water deficit in the dark in a similar direction (Figure 4), which could be further discussed.

3. Pg 12 line 19 refers to bri1-116 bzr1-1D double mutants: “In this line, the BR pathway, BIN2 function within this pathway, is effectively bypassed”. BIN2 activity is not directly affected by the addition of bzr1-D in the double mutant, but BZR1, downstream of BIN2, is constitutively active regardless of the BR signal. Other BIN2 substrates may still be phosphorylated and regulated by BIN2 which would be active in the bri1 background.

4. The authors claim: “In addition, an analysis of root meristem properties in bin2-3bil1bil2 shows that BR signaling is required for the Arabidopsis seedling’s ability to deploy different root growth strategies in response to abiotic stress cues under limiting conditions.” While I agree that bin2-3bil1bil2 shows interesting changes in root morphology and response to combined darkness/water stress, the fact that this mutant has increased BR signaling as the result of loss of BIN2 and homologs (negative regulators) makes it difficult to claim that this genetic analysis indicates a requirement for BR signaling in this context. The loss of function mutants in the pathway (e.g. cpd, bri1-T) would be more appropriate to support the essentiality of BR signaling.

Reviewer #3: The revised manuscript clearly improved. Most comments in the first submission have been addressed and new data was added. Nevertheless, a few issues are listed below:

The focus on brassinosteroids improves the manuscript. However, the mechanistic insights how brassinosteroid are involved in decision making, remain largely obscure. As presented by the data in fig 3 and 4, I would even suggest that the canonical BR-signaling via the BZR1/BES1 signaling is not involved in the “decision-making” as the gain of function mutants bzr1-1D and bes1-D responded like wild type and the bri1-116 bzr1-1D double mutant showed no rescue; BIN2 is known to branch out to integrate other signaling pathways, thus BIN2 possibly operates via one of these pathways. I understand that identifying the mechanism downstream of BIN2 is out of the scope of this work; because of this the claims made on the role of BIN2 should be done very carefully.

There is also an inconsistency on the use of mutants in newly added experiments that should be addressed: hypocotyl cell parameters are from bin2-1, a gain-of-function mutant, while the new data on root meristem and root cell geometry come from bin2-3bil1bil2, a triple loss-of-function mutant. To draw conclusions the gain-of-function and loss of function mutants should be both included.

In Fig. 5d the representative images of bin2-1, at least in my eyes, show a stronger reduction of hypocotyl cell length in darkW compared to dark than shown in the graph of 5e.

The new data on root meristem properties is very interesting.

The summary in Fig. 7 could also be moved into the supplements since the data is presented in Fig. 5 and 6.

**Have all data underlying the figures and results presented in the manuscript been provided?**

Reviewer #1: Yes

Reviewer #2: Yes

Reviewer #3: Yes

PLOS authors have the option to publish the peer review history of their article (what does this mean?). If published, this will include your full peer review and any attached files.

Reviewer #1: **Yes: **Katarzyna Retzer

Reviewer #2: No

Reviewer #3: No

---

## [Editor Report · Decision Letter 1]

23 Nov 2022

Dear Dr Assaad,

We are pleased to inform you that your manuscript entitled "A role for brassinosteroid signalling in decision-making processes in the Arabidopsis seedling." has been editorially accepted for publication in PLOS Genetics. Congratulations!

Before your submission can be formally accepted and sent to production you will need to complete our formatting changes, which you will receive in a follow up email. In addition to those changes, I ask you to change the word ecotypes to accessions, as it is standard in the Arabidopsis community.

Please be aware that it may take several days for you to receive this email; during this time no action is required by you. Please note: the accept date on your published article will reflect the date of this provisional acceptance, but your manuscript will not be scheduled for publication until the required changes have been made.

Yours sincerely,

Claudia Köhler

Section Editor

PLOS Genetics

Comments from the reviewers (if applicable):

**Data Deposition**

http://datadryad.org/submit?journalID=pgenetics&manu=PGENETICS-D-22-01083R1

**Press Queries**

---

## [Editor Report · Acceptance letter]

7 Dec 2022

PGENETICS-D-22-01083R1 

A role for brassinosteroid signalling in decision-making processes in the Arabidopsis seedling. 

Dear Dr Assaad, 

We are pleased to inform you that your manuscript entitled "A role for brassinosteroid signalling in decision-making processes in the Arabidopsis seedling." has been formally accepted for publication in PLOS Genetics! Your manuscript is now with our production department and you will be notified of the publication date in due course.

With kind regards,

Anita Estes

PLOS Genetics

On behalf of:
